# Enhancing the Influence of Labels on Unlabeled Nodes in Graph Convolutional Networks

Jincheng Huang [1]  Yujie Mo [1]  Xiaoshuang Shi [1 2]  Lei Feng [3]  Xiaofeng Zhu [1 4]

## Abstract

The message-passing mechanism of graph convolutional networks (*i.e.,* GCNs) enables label information to reach more unlabeled neighbors, thereby increasing the utilization of labels. However, the additional label information does not always contribute positively to the GCN. To address this issue, we propose a new two-step framework called ELU-GCN. In the first stage, ELU-GCN conducts graph learning to learn a new graph structure (*i.e.,* ELU-graph), which allows the additional label information to positively influence the predictions of GCN. In the second stage, we design a new graph contrastive learning on the GCN framework for representation learning by exploring the consistency and mutually exclusive information between the learned ELU graph and the original graph. Moreover, we theoretically demonstrate that the proposed method can ensure the generalization ability of GCNs. Extensive experiments validate the superiority of our method.

## 1. Introduction

Graph Convolutional Networks (GCNs) (Kipf & Welling, 2017; Gasteiger et al., 2018; Huang et al., 2023a; Xu et al., 2018; Hamilton et al., 2017) have demonstrated remarkable capabilities, primarily due to their ability to propagate label information. This capability has driven their widespread applications in semi-supervised learning. To do this, GCN propagates the representations of unlabeled neighbors to labeled nodes by the message passing mechanism, thereby enabling label information to supervise not only the labeled nodes but also their unlabeled neighbors (Ji et al., 2023;

Dong et al., 2021). Consequently, the framework of optimizing label utilization in GCNs (LU-GCN) has become an increasingly prominent research topic (Wang et al., 2021; Yue et al., 2022; Huang et al., 2024; Yu et al., 2022).

Previous LU-GCN can be partitioned into three categories, *i.e.,* self-training methods, combination methods, and graph learning methods. self-training methods (Dong et al., 2021; Li et al., 2018; Sun et al., 2020; Ji et al., 2023) select unlabeled nodes with the highest classification probability by GCN as training data with pseudo-labels, thus adding the number of labels to improve the GCN. Combination methods (Wang et al., 2021; Yue et al., 2022; Shi et al., 2021) regard the labels as the augmented features so that labels can be used for both representation learning and classification tasks. The feature propagation mechanism allows GCNs to use labels to supervise the representation of both the node itself (*i.e.,* traditional label utilization) and its unlabeled neighbors (*i.e.,* neighboring label utilization). However, the two LU-GCN methods mentioned above primarily focus on optimizing traditional label utilization, neglecting the critical importance of neighboring label utilization in semi-supervised scenarios. Yet, due to noise in the original graph structure, GCNs often struggle to utilize the neighboring labels effectively. To address this issue, recent graph learning methods (Zheng et al., 2020; Luo et al., 2021; Liu et al., 2022) are designed to improve the relationship of every node and its neighbors by updating the graph structure, and thus may potentially improve the neighboring label utilization. For example, Bi et al. (2022) adopt the own and neighbors' label similarity to rewire the graph, which can make features propagate on the same category nodes as possible.

Although existing graph learning methods have achieved promising performance, there are still some limitations that need to be addressed. First, previous methods have used heuristic approaches to learn the graph structure, they have not explored what kind of graph structures can make GCNs effectively propagate the labeled nodes' information to unlabeled nodes. As a result, on the learned graph structure, the label information propagated via message passing may not always contribute positively to the predictions of GCNs. Second, existing graph learning methods fail to explore both the consistency information and the mutually exclusive information between the new graph and the original graph,

[1]School of Computer Science and Engineering, University of Electronic Science and Technology of China, Chengdu, China [2]Sichuan Artificial Intelligence Research Institute, Yibin, 644000, China [3]School of Computer Science and Engineering, Southeast University, Nanjing, China [4]Hainan University, Haikou, China. Correspondence to: Xiaofeng Zhu <seanzhuxf@gmail.com>.

*Proceedings of the $42^{nd}$ International Conference on Machine Learning*, Vancouver, Canada. PMLR 267, 2025. Copyright 2025 by the author(s).

where they have consistent information (*i.e.,* consistency (Xu et al., 2024)), which helps recognize the node effectively, and every graph contains unique and useful information different from another graph, *i.e.,* mutually exclusive information (Wang et al., 2017).

Based on the above observations, a possible solution to improving the effectiveness of GCNs is to define a graph structure that can maximize label utilization during the message-passing process and efficiently combine the original graph. To achieve this, two crucial challenges must be solved, *i.e.,* (i) it is difficult to evaluate whether a graph structure enables GCN to use labels effectively. (ii) it is necessary to mine the consistency and mutually exclusive information between the original graph and the new graph.

In this paper, to address the above issues, different from previous structure improvement methods, we investigate a new framework, *i.e.,* **E**ffectively **L**abel-**U**tilizing GCN (ELU-GCN for brevity), to conduct effective GCN. To achieve this, we first analyze the influence of each class provided by labeled nodes on every unlabeled node. We then optimize the graph structure (*i.e.,* the ELU-graph) by encouraging the label information propagated through the graph to have a positive impact on GCN's prediction results. This ensures that the GCN with the ELU-graph can effectively utilize the label information, thereby addressing **challenge (i)**. Moreover, we address **challenge (ii)** by designing contrasting constraints to bring the consistent information between two graph views (*i.e.,* the original graph and the ELU-graph) closer and push the mutually exclusive information further apart. Finally, we theoretically analyze that the proposed ELU-graph can ensure GCN effectively utilizes labels and improve the generalization ability of the model. Compared with previous methods, our main contributions can be summarized as follows:

- To the best of our knowledge, this is the first work to investigate the limitation of GCNs in effectively utilizing label information under the graph-based learning paradigm. Moreover, we introduce a quantitative framework to analyze which unlabeled nodes struggle to benefit from label supervision by neighborhood.

- We propose an adaptive and parameter-free construction of the ELU-graph to enhance GCNs' ability to utilize label information, particularly for unlabeled nodes. Furthermore, we design a contrastive loss to exploit both the consistency and complementary differences between the ELU-graph and the original graph.

- We theoretically prove that the ELU-graph construction provides generalization guarantees for GCNs. Empirical results on diverse benchmark datasets further validate the superior performance of our method over competitive baselines.

## 2. Related Works

This section briefly reviews the topics related to this work, including GCNs and LPA as well as graph structure learning.

### 2.1. GCNs and LPA

GCNs are the most popular and commonly used model in the field of graph deep learning. Early work attempted to apply the successful convolutional neural network (CNN) to graph structures. For example, CheybNet (Defferrard et al., 2016) first proposes to transform the graph signal from the spatial domain to the spectral domain through the discrete Fourier transform, and then use polynomials to fit the filter shape (*i.e.,* convolution). CheybNet laid the foundation for the development of spectral-domain graph neural networks. The popular GCN was proposed by Kipf et al. (Kipf & Welling, 2017), which is a simplified version of ChebyNet and has demonstrated strong efficiency and effectiveness, thereby promoting the development of the graph deep learning field.

Recently, some works have focused on the combination of LPA and GCN. This is because LPA can characterize the distribution of labels spread on the graph, which can help GCN obtain more category information. Existing combined LPA and GCN methods can be classified into two categories, *i.e.,* pseudo label methods and masked label methods. Pseudo-label methods let the output of LPA serve as the pseudo-labels to guide the representation learning. For example, PTA (Dong et al., 2021) first propagates the known labels along the graph to generate pseudo-labels for the unlabeled nodes, and second, trains normal neural network classifiers on the augmented pseudo-labeled data. GPL (Wu et al., 2024) uses the output of LPA to preserve the edges between nodes of the same class, thereby reducing the intraclass distance. The masked label methods employ LPA as regularization to assist the GCN update parameters or structures. For example, UniMP (Shi et al., 2021) makes some percentage of input label information masked at random, and then predicts it for updating parameters. GCN-LPA (Wang & Leskovec, 2021) also randomly masked a part of the labels and utilized the remaining label nodes to predict them in learning proper edge weights within labeled nodes. Although the above methods achieve excellent results on various tasks, they all overlook whether the additional label information propagated through message passing is effectively utilized by GCNs to improve unlabeled nodes.

### 2.2. Graph Structure Learning

Graph structure learning is an important technology in the graph field. It can improve the graph structure and infer new relationships between samples, thereby promoting the development of graph representation learning or other fields.

Existing Graph structure learning methods can be classified into three categories, *i.e.,* traditional unsupervised graph structure learning methods, supervised graph structure learning methods, and graph rewiring methods.

Traditional unsupervised graph structure learning methods aim to directly learn a graph structure from a set of data points in an unsupervised manner. Early works (Wang & Zhang, 2006; Daitch et al., 2009) exploit the neighborhood information of each data point for graph construction by assuming that each data point can be optimally reconstructed using a linear combination of its neighbors (*i.e.,* $\min_A \|\mathbf{A}\mathbf{X} - \mathbf{X}\|_F^2$). Similarly, (Daitch et al., 2009) introduce the weight (*i.e.,* $\min \sum_i \left\| \mathbf{D}_{i,i}\mathbf{X}_i - \sum_j \mathbf{A}_{i,j}\mathbf{X}_j \right\|^2$). Smoothness (Jiang et al., 2019) is another widely adopted assumption on natural graph signals; the smoothness of the graph signals is usually measured by the Dirichlet energy (*i.e.,* $\min_{\mathbf{A}} \frac{1}{2} \sum_{i,j} \mathbf{A}_{i,j} \|\mathbf{X}_i - \mathbf{X}_j\|^2$ and $\min_{\mathbf{L}} \operatorname{tr}\left(\mathbf{X}^\top \mathbf{L}\mathbf{X}\right)$). The above objective has inspired many advanced graph structure learning methods.

Supervised graph structure learning methods aim to use the downstream task to supervise the structure learning, which can learn a suitable structure for the downstream task. For example, NeuralSparse (Zheng et al., 2020) and PTDNet (Luo et al., 2021) directly use the adjacency matrix of the graph as a parameter and update the adjacency matrix through the downstream task. SA-SGC (Huang et al., 2023b) learns a binary classifier by distinguishing the edges connecting nodes with the same label and the edges connecting nodes with different labels in the training set, thereby deleting the edges between nodes belonging to different categories in the test set. BAGCN (Zhang et al., 2024) uses metric learning to obtain new graph structures and learns suitable metric spaces through downstream tasks.

The goal of graph rewiring methods is to prevent the over-squashing (Alon & Yahav, 2021) problem. For example, FA (Alon & Yahav, 2021) proposed to use a fully connected graph as the last layer of GCN to overcome over-squashing. SDRF (Topping et al., 2022), SJLR (Giraldo et al., 2023), and BORF (Nguyen et al., 2023) aim to enhance the curvature of the neighborhood by rewiring connecting edges with small curvature. They increase local connectivity in the graph topology, indirectly expanding the influence range of labels. Despite their success, existing graph structure learning methods fail to guarantee that the learned structures enable GNNs to propagate label information to unlabeled nodes effectively.

## 3. Method

**Notations.** Given a graph $\mathcal{G} = (V, E, \mathbf{X}, \mathbf{Y})$, where $V$ is the node set and $E$ is the edge set. Original node represen-

tation is denoted by the feature matrix $\mathbf{X} \in \mathbb{R}^{n \times d}$ where $n$ is the number of nodes and $d$ is the number of features for each node. The label matrix is denoted by $\mathbf{Y} \in \mathbb{R}^{n \times c}$ with a total of $c$ classes. The first $m$ points $\mathbf{x}_i (i \leq m)$ are labeled as $\mathbf{Y}_l$, and the remaining $u$ points $\mathbf{x}_i$ ($m + 1 \leq i \leq n$) are unlabeled. The sparse matrix $\mathbf{A} \in \mathbb{R}^{n \times n}$ is the adjacency matrix of $\mathcal{G}$. Let $\mathbf{D} = \operatorname{diag}(d_1, d_2, \cdots, d_n)$ be the degree matrix, where $d_i = \sum_{j \in \mathcal{N}_i} a_{ij}$ is the degree of node $i$, the symmetric normalized adjacency matrix is represented as $\widehat{\mathbf{A}} = \widetilde{\mathbf{D}}^{-\frac{1}{2}} \widetilde{\mathbf{A}} \widetilde{\mathbf{D}}^{-\frac{1}{2}}$ where $\widetilde{\mathbf{A}} = \mathbf{A} + \mathbf{I}$, $\mathbf{I}$ is the identity matrix and $\widetilde{\mathbf{D}}$ is the degree matrix of $\widetilde{\mathbf{A}}$.

### 3.1. Motivation

Given a classification function $f : \mathbf{X} \rightarrow \mathbb{R}^{n \times c}$, the cross entropy losses of Deep Neural Network (DNN) and GCN are formulated by:

$$
\begin{aligned}
\mathcal{L}_{\mathrm{DNN}} &= \mathrm{CE}(f_\theta(\mathbf{X}), \mathbf{Y}) = - \sum_{i \in V_l, k \in C} y_{ik}(\log f_{ik}) \\
\mathcal{L}_{\mathrm{GCN}} &= \mathrm{CE}(\widehat{\mathbf{A}} f_\theta(\mathbf{X}), \mathbf{Y}) = - \sum_{i \in V_l, k \in C} y_{ik}(\log \sum_{j \in \mathcal{N}_i} \widehat{a}_{ij} f_{jk}),
\end{aligned} \tag{1}
$$

where $\theta$ is the parameters of the function $f$. In Eq. (1), the cross entropy loss of DNN is a one-to-one mapping between the feature space and the label space because every label $y_i$ ($l = 1, ..., n$) is only used to supervise the representation learning of one node $v_i$. The mapping $f$ efficiently captures the pattern and distribution of labeled nodes, but it overlooks unlabeled nodes so that the generalization ability of unlabeled nodes is limited. In contrast, the cross entropy loss of the GCN is a one-to-many mapping because its message-passing mechanism can propagate the information from labeled nodes to their neighbors including labeled nodes and unlabeled nodes. As a result, every label $y_i$ is used to supervise the representation learning of both labeled nodes and unlabeled nodes, as shown in the second row of Eq. (1). Hence, unlabeled nodes in the GCN are able to use the label information of labeled nodes to improve the learning of their representations. Obviously, it is critical to ensure that the label information propagated to unlabeled nodes makes a positive contribution to GCN. However, to the best of our knowledge, this issue has not yet received sufficient attention. A related work, GCN-LPA (Wang & Leskovec, 2021), investigates the influence of labeled nodes; however, it primarily focuses on reinforcing mutual influence among labeled nodes, while overlooking their impact on unlabeled nodes. To address this issue, we first quantify how much influence each class, through its labeled nodes, has on the unlabeled nodes.

The recent study in (Xu et al., 2018) reveals that nodes follow the way of random walks to affect other nodes on the graph. Therefore, in this paper, we extend it to obtain the influence of every class of labeled nodes on the unlabeled node by Proposition 3.1, whose proof is provided in Appendix B.1.

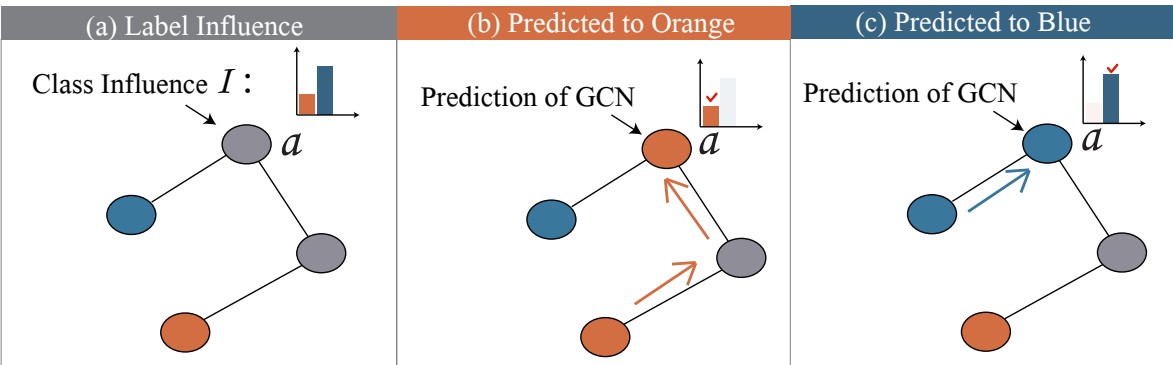

*Figure 1.* An illustration of effective label utilization. Sub-figure (a) wants to assign the label information to node $a$ (gray node) by one unlabeled node (gray node) and two labeled nodes with different classes, *i.e.,* one blue node and one orange node. Moreover, the LPA algorithm is employed to obtain the probability of each labeled node to the node $a$, where the blue node has more influence (or higher probability) than the orange node based on the histogram in the upper right of the sub-figure (a). If the GCN predicts the node $a$ as the orange color (as shown in sub-figure (b)), which is inconsistent with the class with most label information (*i.e.,* blue). It indicates that the label information provided by the message passing of the GCN does not help classify the node $a$, and may even hinder its correct classification. On the contrary, if GCN predicts the node $a$ as the blue color, *i.e.,* sub-figure (c), it implies that the label information provided by the message passing of the GCN helps to classify the node $a$.

**Proposition 3.1.** *Given an unlabeled node $v_i$ ($i = 1, ..., n$), for an arbitrary category $C_l$ ($l = 1, ..., c$), the influence of labeled nodes belong to $C_l$ on the $i$-th node $v_i$ is proportional to the probability that node $v_i$ is classified as $C_l$ by the Label Propagation Algorithm (LPA) in (Zhu, 2005), in the GCN framework.*

Based on Proposition 3.1, LPA can be used to estimate the class-wise probability for unlabeled nodes in the GCN framework. The class with the highest probability is regarded as the most influential, as it contributes the most label information to the node. If this most influential class matches the GCN prediction, it indicates that the label information provided through message passing positively influences the GCN's classification for that node. In this way, the GCN is regarded as having effectively utilized the label information on this node. We provide a case study to illustrate this in Figure 1 and give a formal definition as follows.

**Definition 3.2.** (**Effective label-utilization**) The GCN effectively utilizes label information if the prediction of the GCN is consistent with the output of LPA. This condition is defined at the node level:

$$V_{\text{ELU}} = \{V | \text{LPA}(\mathcal{G}) = \text{GCN}(\mathcal{G})\}, \qquad (2)$$

where $V_{\text{ELU}}$ and $V_{\text{NELU}}$ (*i.e.,* $V_{\text{NELU}} = \{V | \text{LPA}(\mathcal{G}) \neq \text{GCN}(\mathcal{G})\}$), respectively, represent the sets of nodes on which the GCN effectively and ineffectively utilizes label information, respectively.

In real applications, not all unlabeled nodes in GCN frameworks may effectively utilize the label information due to all kinds of reasons, including noise and the distribution of

labeled nodes in the graph. Figure 2 shows that not all unlabeled nodes effectively use label information in the GCN framework (*i.e.,* Figure 2 (a)) and the classification accuracy of $V_{\text{NELU}}$ is lower than that of $V_{\text{ELU}}$ in the same datasets (*i.e.,* Figure 2 (b)). Obviously, NELU nodes influence the effectiveness of the GCN. To address this issue, first, it is crucial to make unlabeled nodes effectively utilize label information. Since label information is propagated through the graph structure. As a result, the graph structure will be updated. Second, the original graph structures often contain noise to influence the message-passing mechanism. Hence, graph learning is obviously a feasible solution.

**3.2. ELU Graph**

Previous graph learning methods generally use either heuristic methods or downstream tasks to conduct graph learning, *i.e.,* updating the graph structure. For example, Pro-GNN (Jin et al., 2020) updates the graph structure through a heuristic approach to constrain the sparsity and smoothness of the graph. PTD-Net (Luo et al., 2021) updates the graph structure by the downstream task, such as the node classification task. However, heuristic methods rely on predefined rules, making it difficult for unlabeled nodes to fully access label-related global information. Downstream task methods focus too much on the performance of labeled nodes, neglecting the role of unlabeled nodes in the graph structure. Therefore, these methods fail to guarantee that the GCN effectively leverages label information for unlabeled nodes. To solve this issue, based on Definition 3.2, we investigate new graph learning methods that ensure unlabeled nodes effectively utilize label information.

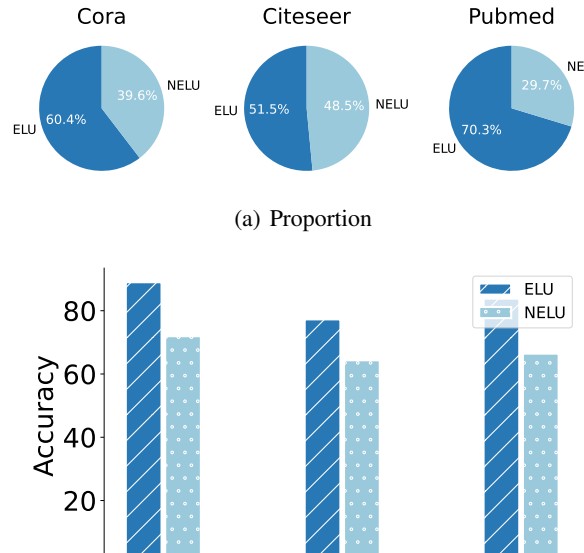

(a) Proportion

(b) Accuracy

*Figure 2.* Visualization of both ELU nodes and NELU nodes in three real datasets, *i.e.,* Cora, Citerseer, and Pubmed. (a) every dataset contains NELU nodes and (b) the classification comparison between ELU nodes and NELU nodes, where ELU nodes have higher classification ability than NELU nodes.

Specifically, denoting the adjacency matrix $\mathbf{S}$ as the ELU graph can ensure the GCN effectively uses the label information. We use Proposition 3.1 to measure the influence of each class on every unlabeled node by the LPA:

$$\mathbf{Q} = \mathbf{SY}, \quad (3)$$

where the $i$-th row of $\mathbf{Q} \in \mathbb{R}^{n \times c}$ (*i.e.,* $\mathbf{Q}_{i,:}$) represents the influence of each class on node $i$. It is noteworthy that $\mathbf{S}$ in Eq. (3) can be the $k$-order of the graph structure. After that, the prediction of GCN with ELU graph can be written as follows (Yang et al., 2023):

$$\hat{\mathbf{Y}} = \mathbf{SH}, \quad s.t. \ \mathbf{H} = \mathrm{MLP}(\mathbf{X}), \quad (4)$$

where $\mathrm{MLP}(\cdot)$ denotes a Multi-Layer Perceptron. Note that the MLP is pre-trained. Therefore, based on Definition 3.2, the ELU graph (*i.e.,* $\mathbf{S}$) can be obtained by minimizing the following objective function:

$$\min \left\| \mathbf{Q} - \hat{\mathbf{Y}} \right\|_F^2 = \min_{\mathbf{S}} \|\mathbf{SY} - \mathbf{SH}\|_F^2. \quad (5)$$

In Eq. (5), the prediction of GCN and the influence of each class are encouraged to be consistent for every node. This item can make all nodes satisfy $\mathrm{LPA}(\mathcal{G}) = \mathrm{GCN}(\mathcal{G})$ in Eq. (2), *i.e.,* this objective function can ensure all nodes can effectively utilize label information by GCN. Therefore,

we can obtain the $\mathbf{S}$ through the optimization algorithm by minimizing the Eq. (5). However, there are some problems with the above objective function. First, it is impractical to solve the above problem directly, as it has a trivial solution: $s_{i,j} = 0, \forall i, \forall j$. Second, LPA generates the prediction for every labeled node to possibly revise the original labels, *i.e.,* the ground truth, adding noisy labels for representation learning. To overcome the above issues, We propose to iteratively update in two steps, *i.e.,* update labels by LPA and update the graph structure $\mathbf{S}$.

In the first step, we calculate the result of LPA $\mathbf{Q}^{(i)}$, *i.e.,* $\mathbf{Q}^{(i)} = \mathbf{S}^{(i-1)}\mathbf{Q}^{(i-1)}$, $(i = 1, \ldots, k)$, where $\mathbf{Q}^{(0)} = \mathbf{Y}$. As a result, Eq.(5) is changed as follows:

$$\min_{\mathbf{S}} \left\| \mathbf{Q}^{(i)} - \mathbf{SH} \right\|_F^2 + \beta \sum_{i,j=1} s_{i,j}^2, s.t. \ \mathbf{Q}_l^{(i)} = \mathbf{Y}_l, \quad (6)$$

where $\beta$ is a non-negative parameter to trade off two terms, the second term can make the subsequent matrix inversion more stable. Eq. (6) holds the closed-form solution to address the first issue. The constraint term "s.t. $\mathbf{Q}_l^{(i)} = \mathbf{Y}_l$" term solves the second issue.

In the second step, we can obtain its closed-form solution as follows:

$$\mathcal{L} = \left\| \mathbf{Q}^{(i)} - \mathbf{SH} \right\|_F^2 + \beta \sum_{i,j=1} s_{i,j}^2$$
$$= Tr((\mathbf{Q}^{(i)} - \mathbf{SH})^T (\mathbf{Q}^{(i)} - \mathbf{SH})) + 2\beta\mathbf{S} \quad (7)$$

where $Tr(\cdot)$ indicates the trace of matrix. Then we have

$$\frac{\partial \mathcal{L}}{\partial \mathbf{S}} = -2\mathbf{Q}^{(i)}\mathbf{H}^T + 2\mathbf{SHH}^T + 2\beta\mathbf{S} \quad (8)$$

Let Eq. (8) equal to 0, we can obtain the closed-form solution $\mathbf{S}^{(i)}$ *i.e.,*

$$\mathbf{S}^{(i)} = \mathbf{Q}^{(i)}\mathbf{H}^T \left( \mathbf{HH}^T + \beta\mathbf{I}_N \right)^{-1}. \quad (9)$$

where $\mathbf{I}_N \in \mathbb{R}^{n \times n}$ is the identity matrix.

Finally, we iteratively optimize Eq. (9) and $\mathbf{Q}^{(i)} = \mathbf{S}^{(i-1)}\mathbf{Q}^{(i-1)}$ to obtain the ELU graph $\mathbf{S}^*$.

However, the calculation of $\mathbf{S}^{(i)}$ in Eq. (9) is with the time complexity of $\mathcal{O}(n^3)$. In this paper, we use the Woodbury identity (Woodbury, 1950) to avoid calculating $\mathbf{S}^{(i)}$ during the iteration process by $\mathbf{Q}^{(i)} = \mathbf{S}^{(i-1)}\mathbf{Q}^{(i-1)}$, *i.e.,*

$$\mathbf{Q}^{(i)} =$$

$$\mathbf{Q}^{(i-1)}\mathbf{H}^T \left( \frac{1}{\beta}\mathbf{I}_N - \frac{1}{\beta^2}\mathbf{H} \left( \mathbf{I}_c + \frac{1}{\beta}\mathbf{H}^T\mathbf{H} \right)^{-1} \mathbf{H}^T \right) \mathbf{Q}^{(i-1)},$$

$$s.t. \ \mathbf{Q}_l^{(i)} = \mathbf{Y}_l, \quad (10)$$

where $\mathbf{I}_c \in \mathbb{R}^{c \times c}$ is the identity matrix and the specific derivation process is listed in the Appendix B.3. Based on the literature (Woodbury, 1950), we can obtain the time complexity of Eq. (10) is $\mathcal{O}(nc^2 + c^3)$, where $c^3 \ll n$. The details are provided in Appendix A.1.

Based on Eq. (10), we obtain $\mathbf{Q}^{(i)}$ $(i = 1, \ldots, k)$ from $\mathbf{Q}^{(i-1)}$. After obtaining $\mathbf{Q}^{(k)}$, we obtain the ELU graph $\mathbf{S}^*$ by calculating Eq. (9) only one time. To achieve efficiency, we employ the Woodbury identity to reduce the time complexity of calculating from cubic to quadratic, *i.e.*,

$$\mathbf{S}^* = \mathbf{Q}^{(k)} (\frac{1}{\beta}\mathbf{H}^T - \frac{1}{\beta^2}\mathbf{H}^T\mathbf{H}\left(\mathbf{I}_c + \frac{1}{\beta}\mathbf{H}^T\mathbf{H}\right)^{-1}\mathbf{H}^T).$$
(11)

The details of Eq. (11) are listed in Section A.2. The pseudocode of calculating Eq. (10) and $\mathbf{S}^*$ is presented in Algorithm 1. In the implementation, we make $\mathbf{S}^*$ sparse by assigning its element less than a threshold as zero, for achieving efficiency. We also use the pseudo labels of ELU nodes to expand the initial $\mathbf{Y}$, for avoiding the issue of limited labels in semi-supervised learning.

---

**Algorithm 1** Pseudo code of calculating $\mathbf{S}^*$.

**Input:** Feature matrix $\mathbf{X}$, label matrix $\mathbf{Y}$, normalized adjacency matrix $\widehat{\mathbf{A}}$, and index of ELU nodes $V_{\text{ELU}}$;
**Output:** ELU graph $\mathbf{S}^*$;
1: $\mathbf{H} = MLP(\mathbf{X})$;
2: Expand initial labels by pseudo labels of ELU nodes;
3: **for** $i \leftarrow 1, 2, \cdots, k$ **do**
4:     Calculate $\mathbf{Q}^{(i)}$ by Eq. (10);
5:     $\mathbf{Q}_l^{(i)} = \mathbf{Y}_l$ in Eq. (10);
6: **end for**
7: Calculate $\mathbf{S}^*$ Eq. (11);
8: **Return** $\mathbf{S}^*$.

---

### 3.3. Contrastive Constraint

Given the ELU graph $\mathbf{S}^*$ and the original graph $\widehat{\mathbf{A}}$, previous graph learning methods often conduct a weighted fusion. For instance, SimP-GCN (Jin et al., 2021) employs a hyperparameter as a weight to fuse the node representation from the original graph with those from the feature similarity graph. However, only performing the weighted sum method may result in incorporating undesirable information from the original graph into the ELU graph. For example, the representation of a NELU node from the original graph might interfere with the learned representation of the corresponding node in the ELU graph. To solve this issue, in this paper, we propose a new contrastive learning paradigm to capture the consistency and mutually exclusive information between these two graphs.

In the ELU graph $\mathbf{S}^*$, all nodes are theoretically ELU nodes. However, the original graph $\widehat{\mathbf{A}}$ includes ELU nodes and

NELU nodes. Obviously, in representation learning, the representations of ELU nodes in both $\mathbf{S}^*$ and $\widehat{\mathbf{A}}$ should be consistent for keeping common information related to the class, the representations of NELU nodes in $\widehat{\mathbf{A}}$ should be different from their representation in $\mathbf{S}^*$.

To do this, we first propose to learn a projection head $p_\theta$ to map both the ELU graph representations and the original graph representations into the same latent space, *i.e.*, $\overline{\mathbf{P}} = p_\theta(\overline{\mathbf{H}})$ and $\widetilde{\mathbf{P}} = p_\theta(\widetilde{\mathbf{H}})$, where $\overline{\mathbf{H}}$ is the representation of the output layer of the GCN dominated by the original graph, and $\widetilde{\mathbf{H}}$ is the representation of the output layer of the GCN dominated by the ELU graph, the output of the model is $\widehat{\mathbf{Y}} = \text{Softmax}((1 - \eta)\overline{\mathbf{H}} + \eta\widetilde{\mathbf{H}})$, where $\eta$ is a hyperparameter. We then design a contrastive loss as follows:

$$\mathcal{L}_{\text{con}} = -\log \frac{\text{Pos}}{\text{Pos} + \text{Neg}} - \frac{1}{n}\sum_{i=1}^{n}\sum_{j=1}^{c} \hat{y}_{i,j} \log \hat{y}_{i,j}$$

$$s.t. \begin{cases} \text{Pos} = \frac{1}{|V_{\text{ELU}}|}\sum_{i=0}^{V_{\text{ELU}}} \exp(d(\overline{\mathbf{P}}_i, \widetilde{\mathbf{P}}_i)/\tau) \\ \text{Neg} = \frac{1}{|V_{\text{NELU}}|}\sum_{j=0}^{V_{\text{NELU}}} \exp(d(\overline{\mathbf{P}}_j, \widetilde{\mathbf{P}}_j)/\tau) \end{cases}$$
(12)

where $d(\cdot)$ is the distance function, $\tau$ denotes the temperature parameter, and $\hat{y}_{i,j}$ is the element in row $i$ and column $j$ in $\widehat{\mathbf{Y}}$.

In Eq. (12), the first term encourages minimizing the distance between every ELU node in the ELU graph and its corresponding node in the original graph, while maximizing the distance between every NELU node in the original graph and its corresponding node in the ELU graph. The second term encourages the decision boundary to be positioned in low-density regions, enhancing the distinction between nodes and ensuring the underlying assumption of semi-supervised learning (Berthelot et al., 2019). As a result, Eq. (12) is available to extract the consistency and mutually exclusive information between the representations dominated by the ELU graph and the original graph.

Finally, the final objective function of our proposed method is obtained by integrating the contrastive loss with the supervised loss (*i.e.*, cross entropy) as follows:

$$\mathcal{L} = CE(\widehat{\mathbf{Y}}, \mathbf{Y}) + \lambda\mathcal{L}_{con}$$
(13)

where $\lambda \in [0, 1]$ is a hyperparameter to fuse the predicted results of two views and two objective functions.

### 3.4. Theoretical Analysis

The ELU graph has been shown to enable the GCN to effectively utilize label information, as demonstrated in Section 3.1. In this section, we theoretically analyze that the generalization ability of the GCN is related to the graph structure and the training labels by Theorem 3.3 (The proof can be found in Appendix B.4):

**Theorem 3.3.** *Given a graph $\mathcal{G}$ with its adjacency matrix $\mathbf{A}$, the label matrix in the training set $\mathbf{Y}$ and the label matrix of the ground truth $\mathbf{Y}_{\text{true}}$, for any unlabeled nodes, if a graph structure makes the labels in training set be consistent to the ground truth, i.e., $\mathbf{Y}_{\text{true}} = \mathbf{A}\mathbf{Y}$, then the upper bound of the generalization ability of the GCN is optimal.*

Based on Theorem 3.3, the graph structure $\mathbf{A}$ maximizes the generalization ability of the GCN if the following equation holds, *i.e.,* $\min_{\mathbf{A}} \|\mathbf{A}\mathbf{Y} - \mathbf{Y}_{\text{true}}\|_F^2$. Therefore, the graph structure can be used to measure if it is suitable for GCN. However, the true labels $\mathbf{Y}_{\text{true}}$ are fixed and unknown. Moreover, the original graph is also fixed so that it is difficult to achieve $\min_{\mathbf{A}} \|\mathbf{A}\mathbf{Y} - \mathbf{Y}_{\text{true}}\|_F^2$. Hence, the original graph should be updated. We then present the following theorem. The proof is listed in Appendix B.5.

**Theorem 3.4.** *The optimization Eq. (5) is equivalent to an approximate optimization of $\min_{\mathbf{A}} \|\mathbf{A}\mathbf{Y} - \mathbf{Y}_{\text{true}}\|_F^2$.*

Theorem 3.4 indicates that the ELU graph can ensure the generalization ability of the GCN.

# 4. Experiments

In this section, we conduct experiments on eleven public datasets to evaluate the proposed method (including citation networks, Amazon networks, social networks, and web page networks), compared to structure improvement methods[1]. Detailed settings are shown in Appendix E. Additional experimental results are shown in Appendix F.

## 4.1. Experimental Setup

### 4.1.1. DATASETS

The used datasets include three benchmark citation datasets (Sen et al., 2008) (*i.e.,* Cora, Citeseer, and Pubmed), two co-purchase networks (Shchur et al., 2018) (*i.e.,* Computers and Photo), two web page networks (Pei et al., 2020) (*i.e.,* Chameleon and Squirrel), which are heterophilic graph data), and four social network datasets (Traud et al., 2012) (*i.e.,* Caltech, UF, Hamilton, and Tulane).

### 4.1.2. COMPARISON METHODS

The comparison methods include three traditional GNN methods, two advanced GNN methods, and seven structure improvement-based GCN methods. Traditional GNN methods include GCN (Kipf & Welling, 2017), GAT (Velickovic et al., 2018), and APPNP (Gasteiger et al., 2018). The advanced GNN methods include GPRGNN (Chien et al., 2021) and PCNet (Li et al., 2024). The structure improvement-based GCN methods include GCN-LPA (Wang & Leskovec,

---

[1]The code is released at https://github.com/huangJC0429/label-utilize-GCN

2021), NeuralSparse-GCN (Zheng et al., 2020), PTDNet-GCN (Luo et al., 2021), CoGSL (Liu et al., 2022), Node-Former (Wu et al., 2022), GSR (Zhao et al., 2023) and BAGCN (Zhang et al., 2024).

### 4.1.3. EVALUATION PROTOCOL

To evaluate the effectiveness of the proposed method, we follow the commonly used setting. Specifically, for the citation network (*i.e.,* Cora, Citeseer, and Pubmed), we use the public split recommended by (Kipf & Welling, 2017) with fixed 20 nodes per class for training, 500 nodes for validation, and 1000 nodes for testing. For Social networks (*i.e.,* Caltech, UF, Hamilton, and Tulane), we randomly generate different data splits with an average train/val/test split ratio of 60%/20%/20%. For the Webpage network (*i.e.,* Chameleon, Squirrel) and co-purchase networks (*i.e.,* Computers, Photo), we use the public splits recommended in the original papers.

## 4.2. Effectiveness Analysis

We first evaluate the effectiveness of the proposed method by reporting the results of node classification in Table 1 and Appendix F, respectively. Obviously, the proposed method obtains better performance on seven datasets than comparison methods.

First, compared with traditional GNN methods and advanced GNN methods, the proposed ELU-GCN outperforms them by large margins on most datasets. For example, the proposed ELU-GCN on average improves by 4.05 %, compared to GCN, and improves by 3.26 % compared to the best advanced GCN method (*i.e.,* PCNet), on all datasets. This demonstrates the superiority of graph structure learning methods, as the label information cannot be effectively utilized for many nodes in the original graph.

Second, compared to the improvement methods, the proposed ELU-GCN achieves the best results, followed by GSR, GCN-LPA, CoGSL, PTDNet-GCN, NeuralSparse-GCN, and NodeFormer. For example, our method on average improves by 2.21% compared to the best comparison method GSR on all seven datasets. This can be attributed to the fact that the proposed ELU-GCN, which can obtain a graph structure (*i.e.,* the ELU graph) that is more suitable for the GCN model to effectively utilize the label information and efficiently mine the consistency and mutually exclusive information between the original graph and the newly obtained graph. In addition, the Webpage networks (i.e., Chameleon and Squirrel) are heterophilic graphs. As mentioned in the theoretical analysis section, the original graph is difficult to guarantee the generalization ability of GCN, especially for heterophilic graphs. Experimental results show that the proposed ELU-GCN outperforms the GCN using the original heterophilic graph by an average of 9.5%, confirming the results of our theoretical analysis.

*Table 1.* Performance on node classification task. The highest results are highlighted in bold. "OOM" denotes out of memory.

| Method | Cora | Citeseer | pubmed | Computers | Photo | Chameleon | squirrel |
|--------|------|----------|--------|-----------|-------|-----------|----------|
| GCN | $81.61_{\pm0.42}$ | $70.35_{\pm0.45}$ | $79.01_{\pm0.62}$ | $81.62_{\pm2.43}$ | $90.44_{\pm1.23}$ | $60.82_{\pm2.24}$ | $43.43_{\pm2.18}$ |
| GAT | $83.03_{\pm0.71}$ | $71.54_{\pm1.12}$ | $79.17_{\pm0.38}$ | $78.01_{\pm19.1}$ | $85.71_{\pm20.3}$ | $40.72_{\pm1.55}$ | $30.26_{\pm2.50}$ |
| APPNP | $83.33_{\pm0.62}$ | $71.80_{\pm0.84}$ | $80.10_{\pm0.21}$ | $82.12_{\pm3.13}$ | $88.63_{\pm3.73}$ | $56.36_{\pm1.53}$ | $46.53_{\pm2.18}$ |
| GPRGNN | $80.55_{\pm1.05}$ | $68.57_{\pm1.22}$ | $77.02_{\pm2.59}$ | $81.71_{\pm2.84}$ | $91.23_{\pm2.59}$ | $46.85_{\pm1.71}$ | $31.61_{\pm1.24}$ |
| PCNet | $82.81_{\pm0.50}$ | $69.92_{\pm0.70}$ | $80.01_{\pm0.88}$ | $81.82_{\pm2.31}$ | $89.63_{\pm2.41}$ | $59.74_{\pm1.43}$ | $48.53_{\pm1.12}$ |
| GCN-LPA | $83.13_{\pm0.51}$ | $72.60_{\pm0.80}$ | $78.64_{\pm1.32}$ | $83.54_{\pm1.41}$ | $90.13_{\pm1.53}$ | $50.26_{\pm1.38}$ | $42.78_{\pm2.36}$ |
| N.S.-GCN | $82.12_{\pm0.14}$ | $71.55_{\pm0.14}$ | $79.14_{\pm0.12}$ | $81.16_{\pm1.53}$ | $89.86_{\pm1.86}$ | $55.37_{\pm1.64}$ | $46.86_{\pm2.02}$ |
| PTDNet-GCN | $82.81_{\pm0.23}$ | $72.73_{\pm0.18}$ | $78.81_{\pm0.24}$ | $82.21_{\pm2.13}$ | $90.23_{\pm2.84}$ | $53.26_{\pm1.44}$ | $41.96_{\pm2.16}$ |
| CoGSL | $81.76_{\pm0.24}$ | $72.79_{\pm0.42}$ | OOM | OOM | $89.63_{\pm2.24}$ | $52.23_{\pm2.03}$ | $39.96_{\pm3.31}$ |
| NodeFormer | $80.28_{\pm0.82}$ | $71.31_{\pm0.98}$ | $78.21_{\pm1.43}$ | $80.35_{\pm2.75}$ | $89.37_{\pm2.03}$ | $34.71_{\pm4.12}$ | $38.54_{\pm1.51}$ |
| GSR | $83.08_{\pm0.48}$ | $72.10_{\pm0.25}$ | $78.09_{\pm0.53}$ | $81.63_{\pm1.35}$ | $90.02_{\pm1.32}$ | $62.28_{\pm1.63}$ | $50.53_{\pm1.93}$ |
| BAGCN | $83.70_{\pm0.21}$ | $72.96_{\pm0.75}$ | $78.54_{\pm0.72}$ | $79.63_{\pm2.52}$ | $\mathbf{91.25_{\pm0.96}}$ | $52.63_{\pm1.78}$ | $42.36_{\pm1.53}$ |
| ELU-GCN | $\mathbf{84.29_{\pm0.39}}$ | $\mathbf{74.23_{\pm0.62}}$ | $\mathbf{80.51_{\pm0.21}}$ | $\mathbf{83.73_{\pm2.31}}$ | $90.81_{\pm1.33}$ | $\mathbf{70.90_{\pm1.76}}$ | $\mathbf{56.91_{\pm1.81}}$ |

Consequently, the effectiveness of the proposed method is verified in node classification tasks.

We further evaluate the effectiveness of the proposed method on social network datasets and report the results of node classification in Appendix F.1. We can observe that the proposed method also achieves competitive results on the social network datasets compared to other baselines. For example, the proposed method outperforms the best baseline (*i.e.,* GSR), on almost all datasets.

### 4.3. Ablation Study

The proposed ELU-GCN framework investigates the ELU graph to enable the GCN to utilize label information effectively. Additionally, a contrastive loss function (*i.e.,* , $\mathcal{L}_{con}$) is introduced to efficiently minimize consistency and mutually exclusive information between the original graph and the ELU graph. To verify the effectiveness of each component of the proposed method and the results are reported in Table 2.

According to Table 2, we can draw the following conclusions. First, our proposed method achieves the best performance when each component is present, indicating that each is essential. This demonstrates the importance of both learning the ELU graph and extracting information from the original graph, as they not only enable GCN to effectively utilize labels but also retain important information in the original graph.

Second, the ELU graph component provided the biggest improvement. For example, the ELU graph improves performance by an average of 2.9% compared to not considering it, and the $\mathcal{L}_{con}$ term improves performance by an average of 1.3% compared to not considering it. This illustrates the

importance of unlabeled nodes being affected by effectively labeled information in message passing.

### 4.4. Visualization

To provide an intuitive and clear understanding of the effectiveness of the learned ELU graph, we visualize the adjacency matrix of the ELU graph in the heatmap on the Cora, Computers, Photo, and Chameleon datasets and report the results in Figure 3.

Specifically, the rows and columns of heatmaps are reordered by node labels. In the heatmaps, the lighter a pixel, the larger the value of the ELU graph matrix weight. From Figure 3, we observe that the heatmaps exhibit a clear block diagonal structure, with each block corresponding to a category. This indicates that the obtained ELU graph tends to increase the weight of connections between nodes of the same category and avoid noisy connections from different classes. As a result, under the GCN framework, the training nodes are likely to propagate the label information to unlabeled nodes of the same category with a high probability, thereby reducing intra-class variance and increasing inter-class distance. Especially on the Chameleon dataset, where the original graph tends to connect nodes with different labels with a high probability (*i.e.,* heterophily). Fortunately, our method can still obtain a graph structure where nodes are connected with the same category, as shown by the experimental results, demonstrating the universality of the proposed method.

### 4.5. Analyze the Trade-off Between Running Time and Accuracy

The biggest limitation of graph structure learning methods is the need to query in $\mathcal{O}(n^2)$ space when learning the adja-

*Table 2.* Classification performance of each component in the proposed method on all datasets.

| Method | Cora | Citeseer | pubmed | Computers | Photo | Chameleon | squirrel |
|---|---|---|---|---|---|---|---|
| GCN | $81.61_{\pm 0.42}$ | $70.35_{\pm 0.45}$ | $79.01_{\pm 0.62}$ | $81.62_{\pm 2.43}$ | $90.44_{\pm 1.23}$ | $60.82_{\pm 2.24}$ | $43.43_{\pm 2.18}$ |
| +ELU graph | $83.49_{\pm 0.55}$ | $72.02_{\pm 0.36}$ | $80.25_{\pm 0.79}$ | $82.56_{\pm 1.23}$ | $90.52_{\pm 1.33}$ | $65.12_{\pm 1.43}$ | $54.12_{\pm 1.32}$ |
| +$\mathcal{L}_{con}$ | $\mathbf{84.29_{\pm 0.39}}$ | $\mathbf{74.23_{\pm 0.62}}$ | $\mathbf{80.51_{\pm 0.21}}$ | $\mathbf{83.73_{\pm 2.31}}$ | $\mathbf{90.81_{\pm 1.33}}$ | $\mathbf{70.90_{\pm 1.76}}$ | $\mathbf{56.91_{\pm 1.81}}$ |

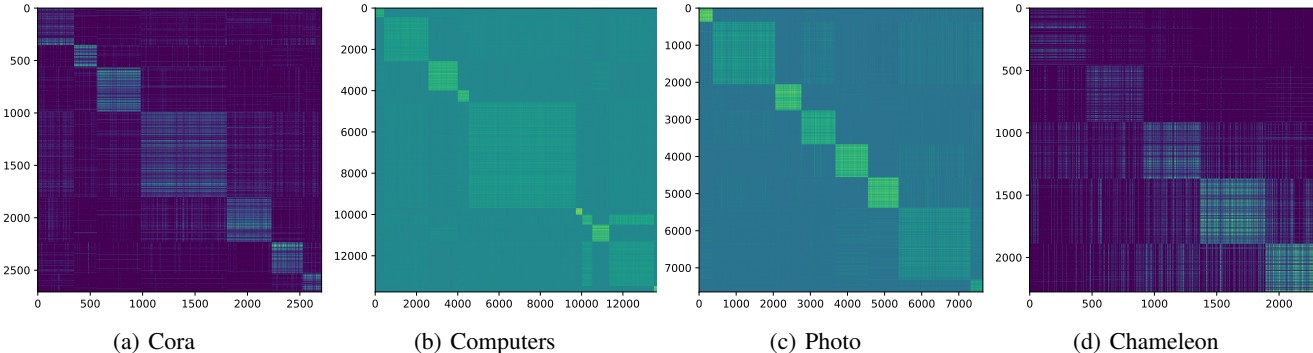

| (a) Cora | (b) Computers | (c) Photo | (d) Chameleon |
|---|---|---|---|

*Figure 3.* Visualization of the adjacency matrix of the ELU graph on Cora, Computers, Photo, and Chameleon datasets. The rows and columns are nodes that are reordered based on node labels, the lighter a pixel, the larger the value of the ELU graph matrix weight.

cency matrix of graph structures. In our proposed method, we cleverly leverage the inverse matrix transformation trick to avoid the computational complexity of $\mathcal{O}(n^2)$ or even higher. Although we have previously analyzed that the complexity of the proposed algorithm in graph construction is $\mathcal{O}(nc^3)$ ($c^3 \ll n$), plus the final graph structure is $\mathcal{O}(n^2)$ (only one calculation is required), we further test the overall actual running time and accuracy of the proposed ELU-GCN and compared with the commonly used baseline (*i.e.,* GCN and GAT). The results are in Figure 4.

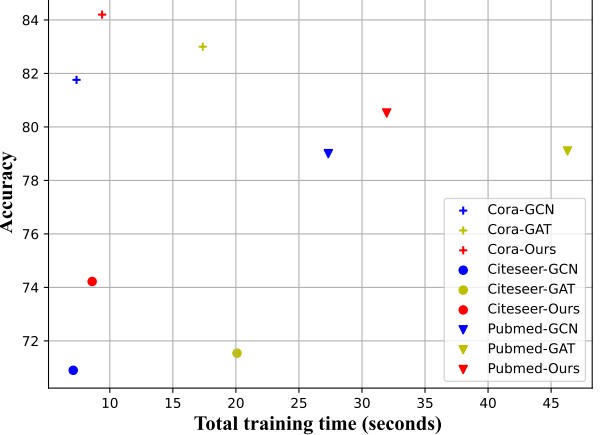

*Figure 4.* Scatter plot showing the relationship between model runtime and accuracy, where the x-axis represents the runtime of different models on different datasets and the y-axis represents their corresponding accuracy (%).

From Figure 4, we have the observations as follows. First, the overall running time of the proposed method is slightly inferior to GCN, but significantly ahead of GAT. This indicates that the proposed ELU graph learning method does not incur too much time overhead and is comparable to the basic GNN model. Second, the proposed method achieves the best classification performance. Combining the above two points, the proposed method achieves the optimal trade-off between running time and model performance.

## 5. Conclusion

In this paper, we study the label utilization of GCN and reveal that a considerable number of unlabeled nodes cannot effectively utilize label information in the GCN framework. Furthermore, we propose a standard for determining which unlabeled nodes can effectively utilize label information in the GCN framework. To enable more unlabeled nodes to utilize label information effectively. We propose an effective label-utilizing graph convolutional network framework. To do this, we optimize the graph structure following the above standard, enabling every unlabeled node to effectively leverage label information. Moreover, we design a novel contrastive loss to capture consistency or mutually exclusive information between the original graph and the ELU graph. Our theoretical analysis demonstrates that ELU-GCN provides superior generalization capabilities compared to conventional GCNs. Extensive experimental results further validate that our method consistently outperforms state-of-the-art methods.

## Acknowledgement

Xiaofeng Zhu was supported in part by the National Key Research and Development Program of China under Grant (No.2022YFA1004100). Xiaoshuang Shi was supported in part by the Sichuan Science and Technology Program under Grant (No. 2024YFHZ0268).

## Impact Statement

This paper presents work whose goal is to advance the field of Machine Learning. There are many potential societal consequences of our work, none of which we feel must be specifically highlighted here.

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

## A. Complexity

### A.1. Complexity of Eq. 10

As mentioned above, by changing the order of matrix multiplication, the time complexity can be reduced, the Eq. 10 is as follows:

$$
\begin{aligned}
\mathbf{Q}^{(i)} &= \mathbf{Q}^{(i-1)} \mathbf{H}^T \left( \frac{1}{\beta} \mathbf{I}_N - \frac{1}{\beta^2} \mathbf{H} \left( \mathbf{I}_c + \frac{1}{\beta} \mathbf{H}^T \mathbf{H} \right)^{-1} \mathbf{H}^T \right) \mathbf{Q}^{(i-1)} \\
&= \mathbf{Q}^{(i-1)} \mathbf{H}^T \left( \frac{1}{\beta} \mathbf{Y} - \frac{1}{\beta^2} \mathbf{H} \left( \mathbf{I}_c + \frac{1}{\beta} \mathbf{H}^T \mathbf{H} \right)^{-1} \mathbf{H}^T \mathbf{Q}^{(i-1)} \right).
\end{aligned}
\tag{14}
$$

We first let $\mathbf{B} = \frac{1}{\beta^2} \mathbf{H} \left( \mathbf{I}_c + \frac{1}{\beta} \mathbf{H}^T \mathbf{H} \right)^{-1} \mathbf{H}^T \mathbf{Q}^{(i-1)}$ and compute it from right to left. Specifically, the matrix inversion operation on a $c \times c$ matrix is $\mathcal{O}(c^3)$. Therefore, the overall time complexity of $\mathbf{S} \in \mathbb{R}^{n \times c}$ is $\mathcal{O}(nc^2 + c^3)$, where $c \ll n$. Then we can compute $\mathbf{Q}^{(i-1)} \mathbf{H}^T \mathbf{B}$, likewise, we calculate it from right to left, this can reduce the time complexity from $\mathcal{O}(n^2 c)$ to $\mathcal{O}(nc^2)$. Therefore the overall time complexity of calculating Eq. 10 is $\mathcal{O}(nc^2 + c^3)$. This significantly improves the model efficiency.

### A.2. Complexity of Eq. 11

Calculating $\mathbf{S}^*$ by eq.(9) will result in $\mathcal{O}(n^3)$ computational cost, which leads to significant memory overhead on large datasets. Thus, we first use the Woodbury identity matrix transformation by Appendix B.3, then the Eq. 9 can be transformed as:

$$
\mathbf{S}^* = \mathbf{Q}^{(i-1)} \mathbf{H}^T \left( \mathbf{H} \mathbf{H}^T + \beta \mathbf{I}_N \right)^{-1} = \mathbf{Q}^{(i-1)} \mathbf{H}^T \left( \frac{1}{\beta} \mathbf{I} - \frac{1}{\beta^2} \mathbf{H} \left( \mathbf{I}_c + \frac{1}{\beta} \mathbf{H}^T \mathbf{H} \right)^{-1} \mathbf{H}^T \right).
\tag{15}
$$

Then, we can transform the calculation order to reduce memory and time overhead as follows:

$$
\begin{aligned}
\mathbf{S}^* &= \mathbf{Q}^{(i-1)} \mathbf{H}^T \left( \frac{1}{\beta} \mathbf{I} - \frac{1}{\beta^2} \mathbf{H} \left( \mathbf{I}_c + \frac{1}{\beta} \mathbf{H}^T \mathbf{H} \right)^{-1} \mathbf{H}^T \right) \\
&= \mathbf{Q}^{(i-1)} \left( \frac{1}{\beta} \mathbf{H}^T - \frac{1}{\beta^2} \mathbf{H}^T \mathbf{H} \left( \mathbf{I}_c + \frac{1}{\beta} \mathbf{H}^T \mathbf{H} \right)^{-1} \mathbf{H}^T \right)
\end{aligned}
\tag{16}
$$

We first let $\mathbf{P} = \frac{1}{\beta^2} \mathbf{H} \mathbf{H}^T \left( \mathbf{I}_c + \frac{1}{\beta} \mathbf{H}^T \mathbf{H} \right)^{-1} \mathbf{H}^T$ and calculate $\mathbf{H}^T \mathbf{H}$, wich time complexity is $\mathcal{O}(nc^2)$, then we can get a $c \times c$ matrix $\mathbf{H}^T \mathbf{H}$, the time complexity of $\left( \mathbf{I}_c + \frac{1}{\beta} \mathbf{H}^T \mathbf{H} \right)^{-1}$ is $\mathcal{O}(c^3)$, thus the overall complexity of $\mathbf{P}$ is $\mathcal{O}(nc^2 + c^3)$. Finally, the complexity of $\mathbf{Q}^{(i-1)} \mathbf{P}$ is $\mathcal{O}(n^2 c)$, since $c$ is the number of classes, it have $c \ll n$. Therefore, the complexity grows quadratically with the number of samples *i.e.,* $\mathcal{O}(n^2)$.

## B. Theoretical Proof

### B.1. Proof for Proposition 3.1

*Proof.* We follow the proof idea of (Wang & Leskovec, 2021), we first introduce a lemma to describe the influence of a node on the other node:

**Lemma B.1.** *(Xu et al., 2018) Assume that the activation function of GCN is* ReLU. *Let* $P_k^{a \to b}$ *be a path* $[v^{(k)}, v^{(k-1)}, \cdots, v^{(0)}]$ *of length* $k$ *from node* $v_a$ *to node* $v_b$, *where* $v^{(k)} = v_a, v^{(0)} = v_b$, *and* $v^{(i-1)} \in \mathcal{N}_{v^{(i)}}$ *for* $i = k, \cdots, 1$. *Then we have the influence of node* $v_a$ *on* $v_b$ *is:*

$$
I(v_b, v_a; k) = \sum_{P_k^{b \to a}} \prod_{i=k}^{1} \tilde{a}_{v^{(i-1)}, v^{(i)}},
\tag{17}
$$

*where $\tilde{a}_{v^{(i-1)},v^{(i)}}$ is the weight of the edge $(v^{(i)}, v^{(i-1)})$.*

The total influence is to sum over all lengths of the path. From Lemma B.1, we can easily obtain the influence of all labeled nodes with label $y_1$ on $v_a$ is

$$I\left(\{v_b : y_v = y_1\}, v_a\right) = \sum_{v_b:y_b=y_1} \sum_{j=1}^{k} \sum_{P_j^{b \to a}} \prod_{i=j}^{1} \tilde{a}_{v^{(i-1)},v^{(i)}}. \tag{18}$$

For LPA, is a random walk algorithm starting from the label node, we denote the classified probability of node $v_a$ in the $y_1$ dimension (*i.e.,* $y_1$ category) as $y_a[y_1]$. It is clear that

$$y_a[y_1] = \frac{y_a[y_1]'}{\sum_{y_i \in y} y_a[y_i]} \quad s.t., \quad y_a[y_1]' = \sum_{v_b:y_b=y_1} \sum_{j=1}^{k} \sum_{P_j^{b \to a}} \prod_{i=j}^{1} \tilde{a}_{v^{(i-1)},v^{(i)}}. \tag{19}$$

Thus, we can get $y_a[y_1] \propto I\left(\{v_b : y_v = y_1\}, v_a\right)$.

$\square$

## B.2. Closed-Form Solution

Given the objective function in Eq. 6, we let

$$\begin{aligned} \mathcal{L} &= \left\|\mathbf{Q}^{(i)} - \mathbf{SH}\right\|_F^2 + \beta \sum_{i,j=1} s_{i,j}^2 \\ &= Tr((\mathbf{Q}^{(i)} - \mathbf{SH})^T(\mathbf{Q}^{(i)} - \mathbf{SH})) + 2\beta\mathbf{S} \end{aligned} \tag{20}$$

where $Tr(\cdot)$ indicates the trace of matrix. Then we have

$$\frac{\partial \mathcal{L}}{\partial \mathbf{S}} = -2\mathbf{Q}^{(i)}\mathbf{H}^T + 2\mathbf{SHH}^T + 2\beta\mathbf{S} \tag{21}$$

Let Eq. 21 equal to 0, we can obtain the closed-form solution $\mathbf{S}^{(i)}$ *i.e.,*

$$\mathbf{S}^{(i)} = \mathbf{Q}^{(i)}\mathbf{H}^T\left(\mathbf{HH}^T + \beta\mathbf{I}_N\right)^{-1}. \tag{22}$$

## B.3. The Woodbury identity

Given four matrices *i.e.,* $\mathbf{A} \in \mathbb{R}^{n \times n}$, $\mathbf{U} \in \mathbb{R}^{n \times k}$, $\mathbf{B} \in \mathbb{R}^{k \times k}$, $\mathbf{V} \in \mathbb{R}^{k \times n}$. We adopt a variation commonly used by the Woodbury identity (Woodbury, 1950) is as follows:

$$(\mathbf{A} + \mathbf{UBV})^{-1} = \mathbf{A}^{-1} - \mathbf{A}^{-1}\mathbf{U}\left(\mathbf{B}^{-1} + \mathbf{VA}^{-1}\mathbf{U}\right)^{-1}\mathbf{VA}^{-1} \tag{23}$$

Without loss of generality, the matrix $\mathbf{A}$ and $\mathbf{B}$ can be replaced with the identity matrix, therefore, we further have

$$(\mathbf{I} + \mathbf{UV})^{-1} = \mathbf{I} - \mathbf{U}(\mathbf{I} + \mathbf{VU})^{-1}\mathbf{V} \tag{24}$$

We can replace the matrices $\mathbf{U}, \mathbf{V}$ with the matrix $\mathbf{H}$ in Eq. 24, thus, we have:

$$\left(\mathbf{HH}^T + \beta\mathbf{I}_N\right)^{-1} = \frac{1}{\beta}\mathbf{I} - \frac{1}{\beta^2}\mathbf{H}\left(\mathbf{I}_c + \frac{1}{\beta}\mathbf{H}^T\mathbf{H}\right)^{-1}\mathbf{H}^T. \tag{25}$$

Therefore, based on Eq. 25, we can transform $\mathbf{Q}^{(i)} = \mathbf{S}^{(i-1)}\mathbf{Q}^{(i-1)}$ as:

$$\begin{aligned} \mathbf{Q}^{(i)} &= \mathbf{S}^{(i-1)}\mathbf{Q}^{(i-1)} \\ &= \mathbf{Q}^{(i-1)}\mathbf{H}^T\left(\frac{1}{\beta}\mathbf{I}_N - \frac{1}{\beta^2}\mathbf{H}\left(\mathbf{I}_c + \frac{1}{\beta}\mathbf{H}^T\mathbf{H}\right)^{-1}\mathbf{H}^T\right)\mathbf{Q}^{(i-1)}. \end{aligned} \tag{26}$$

## B.4. Proof for Theorem 3.3

**Theorem B.2.** *Given a graph $\mathcal{G}$ with adjacency matrix $\mathbf{A}$, training set node label $\mathbf{Y}$ and ground truth label $\mathbf{Y}_{\text{true}}$. For any unknown-label nodes, if $\mathbf{Y}_{\text{true}} = LPA(\mathbf{A}, \mathbf{Y})$, then the upper bound of the GCN's generalization ability reaches optimal on graph $\mathcal{G}$.*

*Proof.* To prove the Theorem 3.3, We first introduce the Complexity Measure to help us understand the generalization ability of GCN. It is the current mainstream method to measure the generalization ability of the model (Neyshabur et al., 2017), which describes the **a lower complexity measure means a better generalization ability**. We follow (Natekar & Sharma, 2020) to adopt Consistency of Representations as our Complexity Measure, which is designed based on the Davies-Bouldin Index (Davies & Bouldin, 1979). Formally, for a given dataset and a given layer of a model, the Davies-Bouldin Index can be written as follows:

$$S_a = \left( \frac{1}{n_a} \sum_{\tau}^{n_a} \left| O_a^{(i)} - \mu_{O_a} \right|^p \right)^{1/p} \quad \text{for } a = 1 \cdots k \tag{27}$$

$$M_{a,b} = \|\mu_{O_a} - \mu_{O_b}\|_p \quad \text{for } a, b = 1 \cdots k, \tag{28}$$

where $a$, $b$ are two different classes, $O_a^{(i)}$ is the GCN smoothed feature of node $i$ belonging to class $a$, $\mu_{O_a}$ is the cluster centroid of the representations of class $a$, here we set $p = 2$, thus $S_a$ measures the intra-class distance of class $a$ and $M_{a,b}$ is a measure of inter-class distance between class $a$ and $b$. Then, we can define complexity measure based on the Davies-Bouldin Index as follows:

$$C = \frac{1}{k} \sum_{i=0}^{k-1} \max_{a \neq b} \frac{S_a + S_b}{M_{a,b}}. \tag{29}$$

We define $P_0$ as the probability that a node's neighbor belongs to the '0-th' class, and $I_0$ as the probability that the node itself belongs to the '0-th' class. Thus, we can calculate the cluster centroid after GCN smoothed features:

$$\mu_{O_0} = \mathbb{E}[O_0^i] = \mathbb{E}[\mathbf{W} \sum_{j \in \mathcal{N}_i} \frac{1}{d_i} \mathbf{X}^j]$$
$$= \mathbf{W}(I_0 P_0 \mu_{X_0} + I_0(1 - P_0)\mu_{X_1}), \tag{30}$$

where $\mathbf{X}^j$ is the 'j-th' node feature and $\mu_{X_i}$ is the cluster centroid of the node features of class $i$. Likewise, we have:

$$\mu_{O_1} = \mathbf{W}(I_1 P_1 \mu_{X_1} + I_1(1 - P_1)\mu_{X_0}). \tag{31}$$

Then, the $M_{0,1}$ can be computed by:

$$\begin{aligned} M_{0,1} &= \|\mu_{O_a} - \mu_{O_b}\| \\ &= \|\mathbf{W}(I_0 P_0 \mu_{X_0} + I_0(1 - P_0)\mu_{X_1} - (I_1 P_1 \mu_{X_1} + I_1(1 - P_1)\mu_{X_0}))\| \\ &= \|\mathbf{W}(I_0 P_0 \mu_{X_0} + I_0 \mu_{X_1} - I_0 P_0 \mu_{X_1} - I_1 P_1 \mu_{X_1} - I_1 \mu_{X_0} + I_1 P_1 \mu_{X_0})\| \\ &= (I_0 P_0 + I_1 P_1) \|\mathbf{W}(\mu_{X_0} - \mu_{X_1})\| + \|I_0 \mu_{X_1} - I_1 \mu_{X_0}\| \\ &\leq (I_0 P_0 + I_1 P_1) \|\mathbf{W}(\mu_{X_0} - \mu_{X_1})\| + \|\mu_{X_1}\| + \|\mu_{X_0}\|. \end{aligned} \tag{32}$$

Then $S_0^2$ is calculated by:

$$\begin{aligned} S_0^2 &= \mathbb{E}\left[ \left\| O_0^{(i)} - \mu_{O_0} \right\|^2 \right] = \mathbb{E}\left[ < O_0^{(i)} - \mu_{O_0}, O_0^{(i)} - \mu_{O_0} > \right] \\ &= \mathbb{E}[(I_0 P_0)(I_0 P_0(X_0 - \mu_{X_0})^T \mathbf{W}^T \mathbf{W}(X_0 - \mu_{X_0}))] \\ &\quad + \mathbb{E}[I_0(1 - P_0)I_0(1 - P_0)(X_1 - \mu_{X_1})^T \mathbf{W}^T \mathbf{W}(X_1 - \mu_{X_1}))] \\ &= I_0^2 P_0^2 \mathbb{E}[\|\mathcal{W}(X_0 - \mu_{X_0})\|] + I_0^2 (1 - P_0)^2 \mathbb{E}[\|\mathcal{W}(X_1 - \mu_{X_1})\|]. \end{aligned} \tag{33}$$

Similarly, we have:

$$
\begin{aligned}
S_1^2 &= \mathbb{E}\left[\left\|O_1^{(i)} - \mu_{O_1}\right\|^2\right] = \mathbb{E}\left[< O_1^{(i)} - \mu_{O_1}, O_1^{(i)} - \mu_{O_1} >\right] \\
&= \mathbb{E}[(I_1 P_1)(I_1 P_1 (X_1 - \mu_{X_1})^T \mathbf{W}^T \mathbf{W}(X_1 - \mu_{X_1}))] \\
&\quad + \mathbb{E}[I_1(1-P_1)I_1(1-P_1)(X_0 - \mu_{X_0})^T \mathbf{W}^T \mathbf{W}(X_0 - \mu_{X_0}))] \\
&= I_1^2 P_1^2 \mathbb{E}[\|\mathcal{W}(X_1 - \mu_{X_1})\|] + I_1^2 (1-P_1)^2 \mathbb{E}[\|\mathcal{W}(X_0 - \mu_{X_0})\|],
\end{aligned}
\tag{34}
$$

where $< \cdot, \cdot >$ is inner production. For simplicity, let $\sigma_0^2 = \mathbb{E}[\|\mathcal{W}(X_0 - \mu_{X_0})\|]$ and $\sigma_1^2 = \mathbb{E}[\|\mathcal{W}(X_1 - \mu_{X_1})\|]$, then the above equation can then be simplified to:

$$
S_0^2 = (I_0 P_0)^2 \sigma_0^2 + (I_0(1-P_0))^2 \sigma_1^2 \geq I_0^2 \frac{\sigma_0^2 \sigma_1^2}{\sigma_0^2 + \sigma_1^2}.
\tag{35}
$$

Similarly, we have:

$$
S_1^2 = (I_1 P_1)^2 \sigma_1^2 + (I_1(1-P_1))^2 \sigma_1^2 \geq I_1^2 \frac{\sigma_0^2 \sigma_1^2}{\sigma_0^2 + \sigma_1^2}.
\tag{36}
$$

Then the complexity measure can be represented as:

$$
C = \frac{\sqrt{S_0^2 + S_1^2 + 2 S_0 \cdot S_1}}{M_{0,1}} \geq \frac{2 \sigma_0 \sigma_1 (I_0 + I_1)^2}{\sqrt{\sigma_0^2 + \sigma_1^2} \cdot ((I_0 P_0 + I_1 P_1) \|\mathbf{W}(\mu_{X_0} - \mu_{X_1})\| + \|\mu_{X_1}\| + \|\mu_{X_0}\|)}.
\tag{37}
$$

Thus, we obtain a lower bound of complexity measure. Also this is the upper bound of the generalization ability. Notice that $\sigma_0$ and $\sigma_1$ could not be zero, otherwise, the classification problem is meaningless. We observe the above equation for nodes with unknown labels and analysis the relationship between the distribution of label $I_0, I_1$ and the lower bound of complexity measure, we find that the probability of their own label (*i.e.,* $I_0$ or $I_1$) and the probability of their neighbors' labels (*i.e.,* $P_0$ or $P_1$) affect the upper bound on their generalization ability. Since $I_0 + I_1 = 1$, we analyze term $(I_0 P_0 + I_1 P_1)$,

$$
(I_0 P_0 + I_1 P_1) = \frac{1}{n} \sum_i^n I_{0,i} P_{0,i} + I_{1,i} P_{1,i}
\tag{38}
$$

where $I_{0,i} \in \{0, 1\}$ is the binary probability that the 'i-th' node label belongs to class 0 where $I_{1,i} = 1 - I_{0,i}$ and $P_{0,i}$ is the probability that the 'i-th' node whose neighbor belongs to class 0. In order to minimize the lower bound of complexity measure, *i.e.,* to maximize the upper bound of generalization ability, it is necessary to maximize $(I_0 P_0 + I_1 P_1)$ here. Obviously, the maximum $(I_0 P_0 + I_1 P_1)$ is obtained at $I_{0,i} = argmax(P_{1,i} P_{0,i})$.

Let's look at the Label Propagation Algorithm(LPA). For nodes with unknown labels,

$$
\widehat{y}_i = \frac{1}{d_i} \sum_{j \in \mathcal{N}_i} y_j.
\tag{39}
$$

Then the probability that the LPA predicts that the 'i-th' node belongs to class 0 can be obtained:

$$
\widehat{I}_{0,i} = argmax\left(\frac{1}{d_i} \sum_{j \in \mathcal{N}_i} y_i == 1, \frac{1}{d_i} \sum_{j \in \mathcal{N}_i} y_i == 0\right) = argmax(P_{1,i} P_{0,i}).
\tag{40}
$$

Similarly, the probability of predicting the 'i-th' node to belong to class 1 is:

$$
\widehat{I}_{1,i} = argmax\left(\frac{1}{d_i} \sum_{j \in \mathcal{N}_i} y_i == 0, \frac{1}{d_i} \sum_{j \in \mathcal{N}_i} y_i == 1\right) = argmax(P_{0,i} P_{1,i}).
\tag{41}
$$

Thus, the upper bound on the generalization ability is maximized when the labels of the unknown label set are distributed as LPA-generated labels.

$\square$

## B.5. Proof for Theorem 3.4

**Theorem B.3.** *The ELU graph can ensure the generalization ability of the GCN, potentially bringing it closer to optimal performance.*

*Proof.* Recall our objective function (*i.e.,* Eq. (5)) $\min_{\mathbf{S}} \|\mathbf{SY} - \mathbf{SH}\|_F^2$, and we first pre-training a GCN (*i.e.,* $\mathbf{SH}$, where $\mathbf{H} = MLP(X)$ is trained in advance) to predict labels for all nodes (*i.e.,* $\widehat{\mathbf{Y}}$), thus our objective function can be rewritten as $\min_{\mathbf{S}} \left\|\mathbf{SY} - \widehat{\mathbf{Y}}\right\|_F^2$, which align with the form of $\min_{\mathbf{A}} \|\mathbf{AY} - \mathbf{Y}_{\text{true}}\|_F^2$ and $\widehat{\mathbf{Y}}$ is often used to estimate $\mathbf{Y}_{\text{true}}$ (Yang et al., 2024; Gong et al., 2023). Therefore, the ELU graph (*i.e.,* $\mathbf{S}$) can ensure the GCN's generalization ability to a certain extent. Moreover, a better adjacency matrix $\mathbf{S}$ can further improve the GCN's predictions (*i.e.,* $\widehat{\mathbf{Y}}$), making $\widehat{\mathbf{Y}}$ increasingly closer to ground truth (*i.e.,* $\mathbf{Y}_{\text{true}}$). Ultimately, we can obtain a graph structure to ensure the GCN's generalization ability is closer to optimal performance. $\square$

# C. Related Works

This section briefly reviews the topics related to this work, including graph convolutional networks, label propagation algorithms, and graph structure learning.

## C.1. GCNs and LPA

Graph convolutional networks (GCNs) are the most popular and commonly used model in the field of graph deep learning. Early work attempted to apply the successful convolutional neural network (CNN) to graph structures. For example, CheybNet (Defferrard et al., 2016) first proposes to transform the graph signal from the spatial domain to the spectral domain through the discrete Fourier transform, and then use polynomials to fit the filter shape (*i.e.,* convolution). CheybNet laid the foundation for the development of spectral-domain graph neural networks. The popular GCN was proposed by Kipf et al. (Kipf & Welling, 2017), which is a simplified version of ChebyNet and has demonstrated strong efficiency and effectiveness, thereby promoting the development of the graph deep learning field.

Recently, some works have focused on the combination of LPA and GCN. This is because LPA can characterize the distribution of labels spread on the graph, which can help GCN obtain more category information. Existing combined LPA and GCN methods can be classified into two categories, *i.e.,* pseudo label methods and masked label methods. Pseudo-label methods let the output of LPA serve as the pseudo-labels to guide the representation learning. For example, PTA (Dong et al., 2021) first propagates the known labels along the graph to generate pseudo-labels for the unlabeled nodes, and second, trains normal neural network classifiers on the augmented pseudo-labeled data. GPL (Wu et al., 2024) uses the output of LPA to preserve the edges between nodes of the same class, thereby reducing the intra-class distance. The masked label methods employ LPA as regularization to assist the GCN update parameters or structures. For example, UniMP (Shi et al., 2021) makes some percentage of input label information masked at random, and then predicts it for updating parameters. GCN-LPA (Wang & Leskovec, 2021) also randomly masked a part of the labels and utilized the remaining label nodes to predict them in learning proper edge weights within labeled nodes. Although the above methods achieve excellent results on various tasks, they fail to explore under what conditions the labels propagated by LPA can best enhance the representation learning of GCNs on unlabeled nodes.

## C.2. Graph Structure Learning

Prior to the recent surge in Graph Neural Networks, the study of graph structure learning had already been extensively explored from multiple perspectives within traditional machine learning.

Graph structure learning is an important technology in the graph field. It can improve the graph structure and infer new relationships between samples, thereby promoting the development of graph representation learning or other fields. Existing Graph structure learning methods can be classified into three categories, *i.e.,* traditional unsupervised graph structure learning methods, supervised graph structure learning methods, and graph rewiring methods. Traditional unsupervised graph structure learning methods aim to directly learn a graph structure from a set of data points in an unsupervised manner. Early works (Wang & Zhang, 2006; Daitch et al., 2009) exploit the neighborhood information of each data point for graph construction by assuming that each data point can be optimally reconstructed using a linear combination of its neighbors (*i.e.,* $\min_A \|\mathbf{AX} - \mathbf{X}\|_F^2$). Similarly, (Daitch et al., 2009) introduce the weight (*i.e.,* $\min \sum_i \left\|\mathbf{D}_{i,i}\mathbf{X}_i - \sum_j \mathbf{A}_{i,j}\mathbf{X}_j\right\|^2$).

Smoothness (Jiang et al., 2019) is another widely adopted assumption on natural graph signals; the smoothness of the graph signals is usually measured by the Dirichlet energy (*i.e.,* $\min_{\mathbf{A}} \frac{1}{2} \sum_{i,j} \mathbf{A}_{i,j} \|\mathbf{X}_i - \mathbf{X}_j\|^2 = \min_{\mathbf{L}} \operatorname{tr}\left(\mathbf{X}^\top \mathbf{L} \mathbf{X}\right)$). Until now, there have been a lot of works based on the above objective function to learn graph structure. Supervised graph structure learning methods aim to use the downstream task to supervise the structure learning, which can learn a suitable structure for the downstream task. For example, NeuralSparse (Zheng et al., 2020) and PTDNet (Luo et al., 2021) directly use the adjacency matrix of the graph as a parameter and update the adjacency matrix through the downstream task. SA-SGC (Huang et al., 2023b) learns a binary classifier by distinguishing the edges connecting nodes with the same label and the edges connecting nodes with different labels in the training set, thereby deleting the edges between nodes belonging to different categories in the test set. BAGCN (Zhang et al., 2024) uses metric learning to obtain new graph structures and learns suitable metric spaces through downstream tasks. The goal of graph rewiring methods is to prevent the over-squashing (Alon & Yahav, 2021) problem. For example, FA (Alon & Yahav, 2021) proposed to use a fully connected graph as the last layer of GCN to overcome over-squashing. SDRF (Topping et al., 2022), SJLR (Giraldo et al., 2023), and BORF (Nguyen et al., 2023) aim to enhance the curvature of the neighborhood by rewiring connecting edges with small curvature. They increase local connectivity in the graph topology, indirectly expanding the influence range of labels. Despite their success, existing graph structure learning methods cannot ensure that the learned graph structure effectively enables GNN models to leverage supervisory information for unlabeled nodes.

# D. Model Detail

## D.1. Pretaining MLP

As mentioned in the method section *MLP has been trained in advance*. Specifically, we employ the two-layer MLP and crosse-entropy to pre-train the MLP:

$$\mathcal{L}_{mlp} : \min_{\Theta_1, \Theta_2} CE(\mathbf{X}\boldsymbol{\Theta}^{(1)}\boldsymbol{\Theta}^{(2)}, \mathbf{Y}) \tag{42}$$

where $\Theta^{(1)}$ and $\Theta^{(2)}$ are learnable parameters. After the above objective function converges by the gradient descent algorithm, we can get $\mathbf{H}$ as follows:

$$\mathbf{H} = \mathbf{X}\boldsymbol{\Theta}^{(1)}\boldsymbol{\Theta}^{(2)}. \tag{43}$$

## D.2. Details of Sparse $\mathbf{S}^*$ and Initialize $\mathbf{Y}$

**Sparse $\mathbf{S}^*$.** $\mathbf{S}^*$ is a fully-connected adjacency matrix. It will bring computationally expensive overhead in message passing, especially for large-scale graph datasets. To mitigate this, we set the elements with small absolute values to 0; Specifically, $\forall i, j$ where $|\mathbf{S}^*_{i,j}| < \eta$, we set $|\mathbf{S}^*_{i,j}| = 0$, while elements with $|\mathbf{S}^*_{i,j}| > \eta$ remain unchanged, where $\eta$ is a non-negative parameter that we usually set to correspond to the top 10 percent of element values. The graph described by $\mathbf{S}^*$ is referred to as the effectively label-utilizing graph (ELU-graph) in this paper.

**Initialize $\mathbf{Y}$.** Since the number of initial label information is very limited in a semi-supervised scenario, having too many rows of all zeros in $\mathbf{Y}$ can cause the algorithm to be unstable. Thus, we propose a label initialization strategy to expand the initial labels with high quality. Specifically, since ELU nodes can effectively utilize the label information and demonstrate high accuracy as shown in Figure 2 (b), we use the pseudo labels of ELU nodes to expand the initial $\mathbf{Y}$.

# E. Experiments Details

## E.1. Datasets

The used datasets include three benchmark citation datasets (Sen et al., 2008) (*i.e.,* Cora, Citeseer, Pubmed), two co-purchase networks (Shchur et al., 2018) (*i.e.,* Computers, Photo), two web page networks (Pei et al., 2020) (*i.e.,* Chameleon and Squirrel, note that these two datasets are heterophilic graph data), and four social network datasets (Traud et al., 2012) (*i.e.,* Caltech, UF, Hamilton, and Tulane). Table 3 summarizes the data statistics. We list the details of the datasets as follows.

- **Citation networks** include Cora, Citeseer, and Pubmed. They are composed of papers as nodes and their relationships such as citation relationships, and common authoring. Node feature is a one-hot vector that indicates whether a word is present in that paper. Words with a frequency of less than 10 are removed.

- **Co-purchase networks** include Photo and Computers, containing 7,487 and 13,752 products, respectively. Edges in

*Table 3.* The statistics of the datasets

| Datasets | Nodes | Edges | Train/Valid/Test Nodes | Features | Classes |
|---|---|---|---|---|---|
| Cora | 2,708 | 5,429 | 140/500/1000 | 1,433 | 7 |
| Citeseer | 3,327 | 4,732 | 120/500/1,000 | 3,703 | 6 |
| Pubmed | 19,717 | 44,338 | 60/500/1,000 | 500 | 3 |
| Amazon Computers | 13,381 | 245,778 | 200/300/12,881 | 767 | 10 |
| Amazon Photo | 7,487 | 119,043 | 160/240/7,084 | 745 | 8 |
| Chameleon | 2,277 | 36,101 | 1,093/729/455 | 2,325 | 5 |
| Squirrel | 5,201 | 217,073 | 2,496/1,665/1,040 | 2,089 | 5 |
| Caltech | 13,882 | 763,868 | 8,240/2,776/2,776 | 6 | 6 |
| UF | 35,123 | 2,931,320 | 21,074/7,024/7,024 | 6 | 6 |
| Hamilton | 2,314 | 192,788 | 1,388/463/463 | 6 | 6 |
| Tulane | 7,752 | 567,836 | 4,652/1,550/1,550 | 6 | 6 |

each dataset indicate that two products are frequently bought together. The feature of each product is bag-of-words encoded product reviews. Products are categorized into several classes by the product category.

- **Webpage networks** include Squirrel and Chameleon, which are two subgraphs of web pages in Wikipedia. Our task is to classify nodes into five categories based on their average amounts of monthly traffic.

- **Social networks** include Caltech, UF, Hamilton, and Tulane, each graph describes the social relationship in a university. Each graph has categorical node attributes with practical meaning (e.g., gender, major, class year.). Moreover, nodes in each dataset belong to six different classes (a student/teacher status flag).

# F. Additional Experiments

## F.1. Node Classification on Social Networks

| Model | Caltech | UF | Hamilton | Tulane |
|---|---|---|---|---|
| GCN | $88.47_{\pm 1.91}$ | $83.94_{\pm 0.61}$ | $92.26_{\pm 0.35}$ | $87.93_{\pm 0.97}$ |
| GAT | $81.17_{\pm 2.15}$ | $81.68_{\pm 0.59}$ | $91.43_{\pm 1.25}$ | $84.45_{\pm 1.45}$ |
| APPNP | $90.76_{\pm 2.38}$ | $83.07_{\pm 0.54}$ | $93.29_{\pm 0.47}$ | $88.52_{\pm 0.44}$ |
| GCN-LPA | $89.12_{\pm 2.11}$ | $83.78_{\pm 0.69}$ | $92.56_{\pm 0.87}$ | $88.32_{\pm 1.02}$ |
| GSR | $90.23_{\pm 2.41}$ | $84.01_{\pm 0.63}$ | $92.45_{\pm 0.84}$ | $88.75_{\pm 1.01}$ |
| Ours | $\mathbf{91.93_{\pm 0.69}}$ | $\mathbf{85.62_{\pm 0.53}}$ | $\mathbf{93.65_{\pm 0.78}}$ | $\mathbf{89.30_{\pm 0.77}}$ |

We further evaluate the effectiveness of the proposed method on the social network datasets by reporting the results of node classification. Obviously, our method achieves the best effectiveness on node classification tasks.

Specifically, the proposed method achieves competitive results on the social network datasets compared to other baselines. For example, the proposed method on average improves by 1.27%, compared to the best baseline (i.e., GSR), on almost all datasets. This demonstrates the universality of our method, which can achieve excellent results in most datasets.

## F.2. Parameter Analysis

In the proposed method, we employ the non-negative parameters (*i.e.,* $\lambda$, $\tau$, and $eta$) to achieve a trade-off between the supervised loss and the contrastive loss, the temperature control, and the fusion of the ELU graph and the original graph. To investigate the impact of $\lambda$, $\tau$, and $\eta$ with different settings, we conduct the node classification on the Cora and Citeseer datasets by varying the value of $\lambda$, $\tau$, and $\eta$ in the range of [0.1, 1.0]. Note that the smaller the $\tau$, the closer the model brings the positive samples and the further apart the negative samples. The results are reported in Figure 5 and 6, respectively.

From Figure 5, we have the following observations: First, the proposed method achieves significant performance when the parameter $\tau$ is in the range of [0.1, 0.3] if $\tau$ values are too large (e.g., $\tau > 0.3$), the performance degrades. For $\tau$, setting it

to 1 is equivalent to not using the temperature coefficient. The lower the temperature coefficient, the stronger the effect, indicating that $\tau$ is essential for the proposed method. Note that $\tau$ cannot be set to 0 because it is the denominator. Second, when $\lambda$ is set in the range of [0.3, 0.6], the model can get a higher performance, if the value of $\lambda$ is too high or too small (e.g., =0, the results shown in the Table 2), the performance degrades. This indicates that the proposed contrastive loss is necessary for the model.

From Figure 6, for the parameter $\eta$, the proposed method achieves the best results while the value of the parameter is set in the range of [0.1,0.5]. This further confirms the importance of both the ELU graph and the original graph.

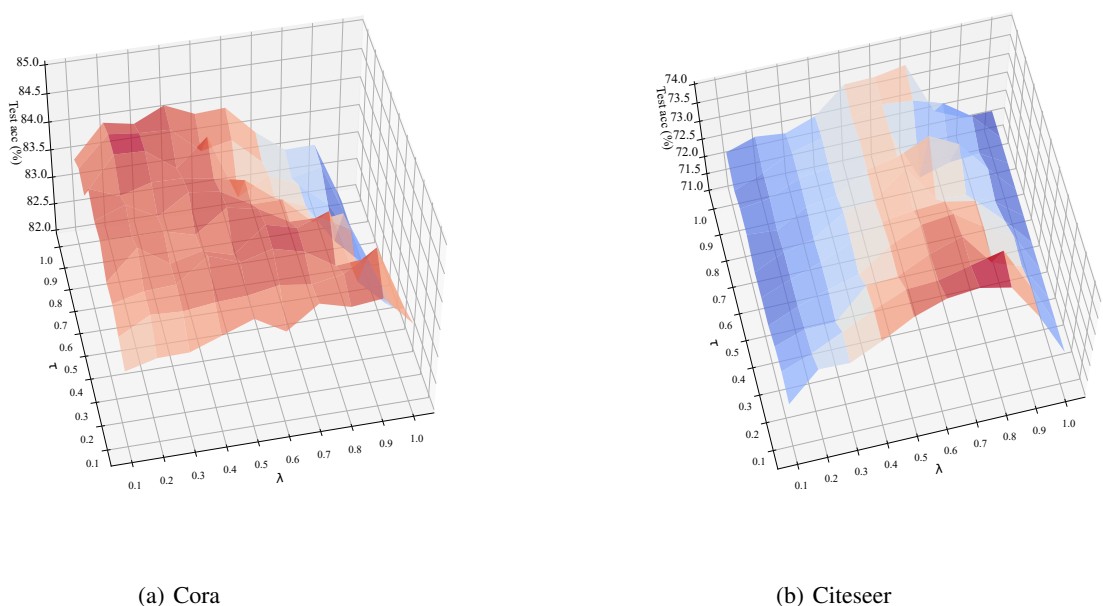

(a) Cora                                                         (b) Citeseer

*Figure 5.* The classification performance of the proposed method at different parameter settings (*i.e.,* $\tau$, $\lambda$) on the Cora and Citeseer datasets.

### F.3. Analysis the Improve on $V_{\text{NELU}}$

To better examine the effectiveness of the proposed ELU-GCN on $V_{\text{NELU}}$, we further evaluate the model's improvement over GCN on $V_{\text{NELU}}$ on Cora, Citeseer, and Pubmed datasets. The results are shown in Figure 7.

Specifically, the proposed ELU-GCN shows a particularly significant improvement on $V_{\text{NELU}}$ across the three datasets. For example, our method on average improves by 3.7 % on $V_{\text{NELU}}$ and 2.1% on all test nodes compared to GCN on these three datasets. This can be attributed to the fact that the proposed ELU-GCN provides the ELU graph that can make $V_{\text{NELU}}$ utilize the label information more effectively under the GCN framework, and this also indicates that the main improvement of the proposed ELU-GCN is on $V_{\text{NELU}}$.

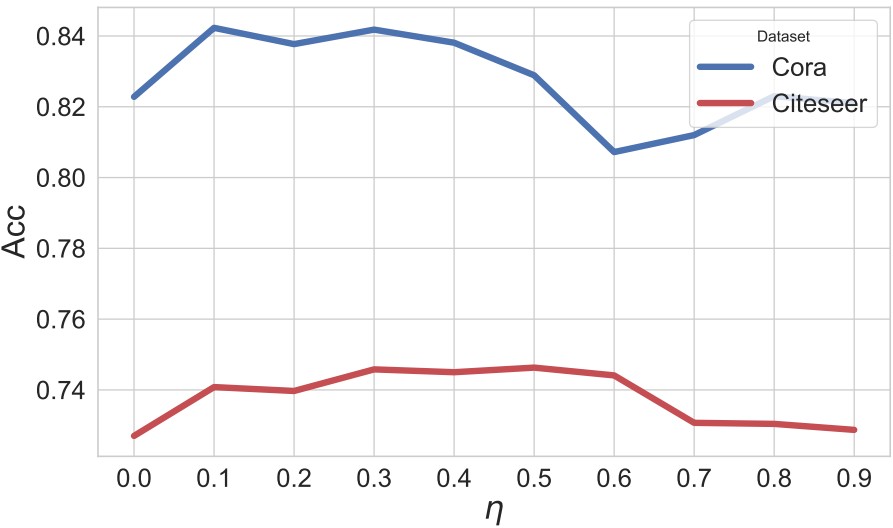

*Figure 6.* The classification performance of the proposed method at different parameter settings (*i.e.*, $\eta$) on the Cora and Citeseer datasets.

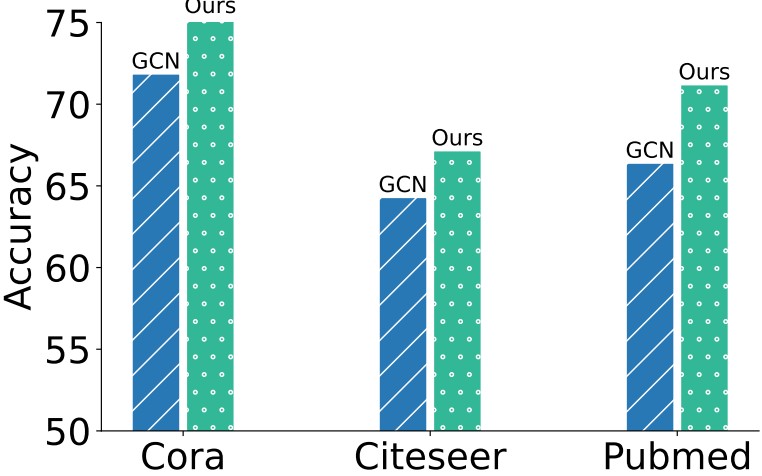

*Figure 7.* The accuracy of ELU-GCN and GCN of $V_{\mathrm{NELU}}$ on Cora, Citeseer, and Pubmed datasets.

