# OpenReview forum: "Enhancing the Influence of Labels on Unlabeled Nodes in Graph Convolutional Networks"
_ICML.cc/2025/Conference — ICML 2025 poster_

### Official Review · Reviewer_sZPZ · 2025-03-06

**Overall Recommendation:** 4

**Summary:**

The paper focuses on the impact of labels on unlabeled nodes, the authors propose that the label information is not always effectively utilized in the traditional GCN framework. Thus, this paper proposes the ELU-GCN to solve this issue. First, the paper proposed a new objective function to ensure the graph structure effectively propagates label information under the GCN framework. Second, the paper proposes new contrastive learning to capture consistent and mutually exclusive information between the two graphs.

**Claims And Evidence:**

- The experimental results demonstrate that ELU-GCN outperforms existing methods on multiple datasets, particularly on heterophilic graphs like Chameleon.
- The motivation is reasonable.
- Theoretical proof is basically reasonable.

**Essential References Not Discussed:**

- The essential references are discussed.

**Experimental Designs Or Analyses:**

Strength:
- The experimental setup and dataset selection are reasonable.

Weakness:
- Lack of baselines. Since the authors designed a new contrastive learning method, it is necessary to compare it with the contrastive learning method.

**Methods And Evaluation Criteria:**

- The proposed two-stage method is clear and promising.
- The Woodbury identity trick effectively reduces the computational complexity of ELU-graph, making the framework more efficient.
- The paper uses 11 public datasets, covering both homophilic and heterophilic graphs, making the evaluation fairly comprehensive.

**Other Comments Or Suggestions:**

- There are some minor errors that need to be carefully checked, such as whether the comma or period at the end of the formula is correct.
- The paper directly mentions LPA and GCN, but there is no introduction to them. The author should briefly introduce them in the notation or appendix.

**Other Strengths And Weaknesses:**

Other Strengths:
- The issue the paper focuses on is interesting i.e., Enhancing the use of label information by unlabeled samples.
- Overall, the paper is complete and inspiring.

Other Weaknesses:
- The proposed Eq. (12) is a paradigm of contrastive learning. The author should further discuss its difference and connection with contrastive learning loss such as InfoNCE.
- The proposed method seems a bit complicated, therefore I am concerned about its reproducibility.

**Questions For Authors:**

See weaknesses above.

**Relation To Broader Scientific Literature:**

Strength:
- The paper provides a well-structured overview of LU-GCN and the differences and advantages over previous methods are also explained.

Weakness:
- The discussion on contrastive learning in graphs is somewhat lacking.

**Theoretical Claims:**

- The theoretical analysis appears mathematically sound.

---

> ### Author Rebuttal · Authors · 2025-03-30
>
> Thanks for the positive comments. We are so encouraged and will try our best to address the concerns one by one. All changes here will be found in the final version.
>
> >Q1: Lack of baselines. Since the authors designed a new contrastive learning method, it is necessary to compare it with the contrastive learning method.
>
> A1: We acknowledge the importance of comparing our method with existing contrastive learning methods. To address this, we compared the typical graph contrastive learning GRANCE[1] and the recent graph contrastive learning method (SGCL[2]). We can observe that our method consistently outperforms the two contrastive learning baselines across all datasets.
>
> [1] Deep graph contrastive representation learning. ICML'20
>
> [2] Rethinking and Simplifying Bootstrapped Graph Latents. WSDM'24
>
> | Datasets | Cora           | Citeseer       | pubmed         | Computers      | Photo          | Chameleon      | squirrel       |
> |----------|----------------|----------------|----------------|----------------|----------------|----------------|----------------|
> | GRACE    | 83.30±0.40     | 72.10±0.50     | 79.86±0.12     | 81.86±3.86     | 88.72±2.07     | 46.75±2.47     | 38.16±3.14     |
> | SGCL     | 83.54±0.40     | 72.58±0.25     | 80.09±0.53     | 81.67±2.59     | 89.07±2.45     | 55.36±1.28     | 42.34±1.47     |
> | ELU-GCN  | **84.29±0.39** | **74.23±0.62** | **80.51±0.21** | **83.73±2.31** | **90.81±1.33** | **70.90±1.76** | **56.91±1.81** |
>
> >Q2: The discussion on contrastive learning in graphs is somewhat lacking. The proposed Eq. (12) is a paradigm of contrastive learning. The author should further discuss its difference and connection with contrastive learning loss, such as InfoNCE.
>
> A2: A key difference is that in InfoNCE, the same node in different graph views is treated as a positive sample, while different nodes are treated as negative samples. In contrast, Eq. (12) distinguishes ELU nodes and NELU nodes, ensuring that ELU nodes maintain consistency across graphs while NELU nodes are pushed apart. This means that Eq. (12) incorporates structural information specific to label influence in GCNs, rather than relying solely on node (instance) discrimination as in InfoNCE. We will clarify this point in the revised version.
>
> >Q3: The proposed method seems a bit complicated, therefore I am concerned about its reproducibility.
>
> A3: In fact, our method is simple to implement in practice. The computation of the ELU-graph is a parameter-free process, and the Woodbury trick significantly accelerates matrix operations. Moreover, we have already provided the code link in the supplementary material and commit to open-sourcing it to ensure full reproducibility.
>
> >Comment: There are some minor errors that need to be carefully checked, such as whether the comma or period at the end of the formula is correct.
>
> Thank you for your careful review. We will thoroughly check and correct any minor errors, including punctuation issues in the formulas.
>
> >Comment: The paper directly mentions LPA and GCN, but there is no introduction to them. The author should briefly introduce them in the notation or appendix.
>
> Thank you for your suggestion. We will add brief introductions to LPA and GCN in the notation section or appendix to improve clarity.

---

### Official Review · Reviewer_Dm6W · 2025-03-09

**Overall Recommendation:** 3

**Summary:**

This paper introduces ELU-GCN which enhances label utilization in GCNs. First, it constructs an ELU graph to optimize label influence on unlabeled nodes. Then, a contrastive loss is designed to enhance representation learning by integrating information from both the ELU graph and the original graph. The experiments on multiple datasets show its superiority over existing approaches.

**Claims And Evidence:**

The claims in the paper are supported by both experimental and theoretical evidence. The authors have validated the effectiveness of the proposed method on multiple datasets. However, some of the assumptions in the paper may be problematic. Specifically, the core idea is to ensure that the prediction of GCN is consistent with the output of LPA. However, this may be problematic—Why not constrain the GCN output to align with the LPA during training? Moreover, this approach does not seem to be effective.

**Essential References Not Discussed:**

No

**Experimental Designs Or Analyses:**

The authors verified the effectiveness of the method on multiple benchmark datasets and conducted detailed comparative experiments and statistical analysis. However, some details of the experimental settings are missing, such as learning rate, weight decay, number of hidden units, etc.

**Methods And Evaluation Criteria:**

The proposed method is feasible, and its effectiveness has been validated on a total of 11 datasets.

**Other Comments Or Suggestions:**

Please check the citation format and distinguish the use of \cite and \citet. For example, in the last sentence of the second paragraph of the Introduction, Bi et al should use \citet instead of \cite.

**Other Strengths And Weaknesses:**

Besides the strengths and weaknesses mentioned above, the paper has the following additional strengths and weaknesses.

Other Strengths:
- The paper is well-written.
- The experiment is quite sufficient.

Other weaknesses:
- The subscript notation of $\mathbf{Y}$ is confusing. In Notation, $\mathbf{Y}_{l}$ is defined as the training set label, and then becomes $\mathbf{Y}$ in Eq. (13) and Theorem 2.3. Please check and unify.
- The S* calculated by Eq. (11) looks like a dense matrix (almost no zeros), so it consumes a lot of memory and time for subsequent matrix multiplication.
- The authors mention heterophily graphs but do not give any definition.

**Questions For Authors:**

I would like to ask whether this framework can be used for GAT.

**Relation To Broader Scientific Literature:**

The key contributions of the paper are closely related to the existing literature, such as GCN-LPA and graph contrastive learning. Building on a thorough citation of relevant research, the authors propose a new method that addresses the issue of label information not being effectively utilized by GCNs.

**Theoretical Claims:**

The proof of the theory seems to be sound.

---

> ### Author Rebuttal · Authors · 2025-03-30
>
> Thanks for the positive comments. We are so encouraged and will try our best to address the concerns one by one. All changes here will be found in the final version.
>
> >Q1: The core idea is to ensure that the prediction of GCN is consistent with the output of LPA. However, this may be problematic—Why not constrain the GCN output to align with the LPA during training? Moreover, this approach does not seem to be effective.
>
> A1: First, we would like to clarify that our method does not simply force GCN predictions to align with LPA; rather, LPA outputs should also align with GCN predictions. Therefore, directly constraining the GCN output to match LPA is inappropriate. Instead, our approach aims to find a new graph structure that naturally ensures consistency between GCN and LPA outputs, allowing the prediction of GCN to utilize label information effectively (see Section 2.1).
>
> >Q2: Some details of the experimental settings are missing, such as learning rate, weight decay, number of hidden units, etc.
>
> A2: We set the weight decay to
> 5e-4. The learning rate was selected from [0.01, 0.02] for all datasets, and the number of hidden units was chosen from [4, 8, 64, 128]. We will include these details in the revised version of the paper to ensure clarity and completeness.
>
> >Q3: The subscript notation of $\mathbf{Y}$ is confusing. In Notation, $\mathbf{Y}_{l}$ is defined as the training set label, and then $\mathbf{Y}$ becomes in Eq. (13) and Theorem 2.3. Please check and unify.
>
> A3: Thank you for your careful observation. We acknowledge the inconsistency in the subscript notation of
> $\mathbf{Y}$. In the notation section,
> $\mathbf{Y} _{l}$ refers to the training set labels, and the same meaning should be maintained in Eq. (13) and Theorem 2.3. We will revise the notation to ensure consistency and clarity in the next version.
>
> >Q4: The $\mathbf{S}^{*}$ calculated by Eq. (11) looks like a dense matrix (almost no zeros), so it consumes a lot of memory and time for subsequent matrix multiplication.
>
> A4: You are correct that
> $\mathbf{S}^*$ computed from Eq. (11) is a dense matrix, which could lead to high memory and computational costs for subsequent matrix multiplications. However, we apply a sparsification process to  $\mathbf{S}^*$ to mitigate this issue. Specifically, $ \forall i,j$ where $ |\mathbf{S}^* _{i,j} | < \eta$, we set $|\mathbf{S}^* _{i,j} | = 0$, while elements with $ |\mathbf{S}^{*} _{i,j} | > \eta$ remain unchanged, where $\eta$ is a non-negative parameter that we usually set to correspond to the top 10 percent of element values. The details of this process are already provided in the appendix.
>
> >Q5: The authors mention heterophily graphs but do not give any definition.
>
> A5: We acknowledge the omission and will provide a clear definition of heterophily graphs in the revised version. Generally, heterophily graphs refer to graphs where connected nodes tend to have different labels, in contrast to homophily graphs where connected nodes are more likely to share the same label.
>
> >Q6: I would like to ask whether this framework can be used for GAT.
>
> A6: Although our method is specifically designed for GCN, we have added experiments applying the ELU graph to GAT.
>
> | Datasets | Cora           | Citeseer       | Pubmed         | Computers      | Photo          | Chameleon      | Squirrel       |
> |----------|----------------|----------------|----------------|----------------|----------------|----------------|----------------|
> | GAT      | 83.03±0.71     | 71.54±1.12     | 79.17±0.38     | 78.01±19.1     | 85.71±20.3     | 40.72±1.55     | 30.26±2.50     |
> | ELU-GAT  | **84.89±0.39** | **74.53±0.49** | **80.23±0.41** | **80.36±5.36** | **88.43±1.85** | **60.17±2.14** | **50.47±1.04** |
>
>
> The results show that the ELU graph also positively impacts GAT, demonstrating its effectiveness for GAT as well.

---

> > ### Comment · Reviewer_Dm6W · 2025-04-03
> >
> > Thanks for the author's rebuttal, which addressed most of my concerns. I am happy to raise my rating from 2 to 3.

---

### Official Review · Reviewer_vmVu · 2025-03-10

**Overall Recommendation:** 3

**Summary:**

This paper proposed a new GCN framework called ELU-GCN, which aims to better propagate the label information to unlabeled nodes. First, by analyzing which situation can achieve effective label utilization for unlabeled nodes, the authors proposed an objective function that can guide the GCN to effective label utilization. Then, a graph regularization is designed to capture the consistency and mutually exclusive information between the original and the ELU graphs.

**Claims And Evidence:**

The majority of the claims are well-supported by empirical results and theoretical analysis.

**Essential References Not Discussed:**

None.

**Experimental Designs Or Analyses:**

I have reviewed the experimental design and analyses, and while they are generally sound, there are some areas that could be improved. For example, the real SOTA GNN mode such as GCNII[1] needs to be compared.
[1] Chen M, Wei Z, Huang Z, et al. Simple and deep graph convolutional networks.ICML.

**Methods And Evaluation Criteria:**

The proposed ELU-GCN makes sense for the GNN field. The selected datasets include both homophilic and heterophilic graphs, making the evaluation relatively comprehensive.

**Other Comments Or Suggestions:**

See weaknesses.

**Other Strengths And Weaknesses:**

Strengths
1. This paper is well-written and clearly structured.

2. The proposed ELU-GCN is novel.

Weaknesses:

1. Figure 2 gives the experimental conclusion very abruptly. The specific experimental details of Figure 2 need to be supplemented.

2. There are many hyperparameters in this paper. Although the author has analyzed the sensitivity of some hyperparameters, some important parameters have not been analyzed, such as the number of iterations used to calculate the ELU graph.

3. In Eq. 12, it is necessary to explain what specific distance function is used and
why it is chosen.

**Questions For Authors:**

From Theorem 2.3, can this also be understood as: if an adjacency matrix A makes LPA (AY) perform better, then it would also be better for GCN? Would this inspire the use of a more lightweight and parameter-free LPA instead of GCN to find a better graph structure?

**Relation To Broader Scientific Literature:**

GCN-LPA also studies the impact of labels, but the author claims the differences and Strengths of GCN-LPA in the paper.

**Theoretical Claims:**

This paper provides additional theoretical analysis, and the theoretical proof is complete and reasonable.

---

> ### Author Rebuttal · Authors · 2025-03-30
>
> Thanks for the positive comments. We are so encouraged and will try our best to address the concerns one by one. All
> changes here will be found in the final version.
>
> >Q1: The real SOTA GNN mode such as GCNII[1] needs to be compared. [1] Chen M, Wei Z, Huang Z, et al. Simple and deep graph convolutional networks.ICML.
>
> A1: We have added a comparison with GCNII in the updated experiments as following table. The results show that, on the Cora dataset, GCNII performs a little bit better, but on the other six datasets, our method outperforms GCNII. We will include these results in the next version for further clarification.
>
> | Datasets | Cora           | Citeseer       | pubmed         | Computers      | Photo          | Chameleon      | squirrel       |
> |----------|----------------|----------------|----------------|----------------|----------------|----------------|----------------|
> | GCNII    | **85.49±0.52** | 73.41±0.63     | 80.28±0.41     | 82.53±4.02     | 87.48±2.14     | 62.48±2.54     | 48.17±2.04     |
> | ELU-GCN  | 84.29±0.39     | **74.23±0.62** | **80.51±0.21** | **83.73±2.31** | **90.81±1.33** | **70.90±1.76** | **56.91±1.81** |
>
> >Q2: Figure 2 gives the experimental conclusion very abruptly. The specific experimental details of Figure 2 need to be supplemented.
>
> A2: Thank you for your suggestion. The experimental setup for Figure 2 follows the same configuration outlined in Section 3. The specific details of the experimental setup can be found in the appendix. We will enhance the explanation of this section in the next version to provide more clarity.
>
> >Q3: In Eq. 12, it is necessary to explain what specific distance function is used and why it is chosen.
>
> A3: In Eq. 12, various distance functions, such as Euclidean distance and inner product, can be considered. Based on experimental results, we found that Euclidean distance performed better in our case.
>
> >Q4: From Theorem 2.3, can this also be understood as: if an adjacency matrix A makes LPA (AY) perform better, then it would also be better for GCN? Would this inspire the use of a more lightweight and parameter-free LPA instead of GCN to find a better graph structure?
>
> A4: Yes, I agree with your opinion. Based on Theorem 2.3, if a graph structure $\mathbf{A}$ enables LPA to achieve better performance, it means that GCN can achieve better generalization ability on this graph structure $\mathbf{A}$. This suggests that a lightweight and parameter-free LPA can serve as a criterion for evaluating graph structures or even as an objective function in graph structure learning. In fact, our method can also be seen as optimizing the graph structure $\mathbf{A}$ to improve LPA performance as much as possible.

---

### Official Review · Reviewer_3mCS · 2025-03-11

**Overall Recommendation:** 3

**Summary:**

The paper proposes ELU-GCN, a two-stage method. First, it constructs the ELU graph, which enables the message passing in GCN to utilize label information more effectively. After that, a contrastive loss is designed to fuse information between the original graph and the ELU graph.

**Claims And Evidence:**

The paper claims that ELU-GCN enhances label utilization in GCNs through adaptive graph construction and contrastive learning, with experimental results and the paper’s analysis generally supporting this claim.

**Essential References Not Discussed:**

As far as I know, no essential references have been overlooked.

**Experimental Designs Or Analyses:**

The experimental design is reasonable, validating the proposed model from the perspectives of effectiveness, ablation study, runtime analysis, etc. Notably, the visualization of the key ELU graph helps provide a clearer understanding of the ELU graph.

**Methods And Evaluation Criteria:**

The proposed method is based on mathematical derivations and effectively addresses complexity issues encountered in the process, making it a reasonable approach. The evaluation criteria are appropriate for studies in this field.

**Other Comments Or Suggestions:**

Please reply or modify according to the Weaknesses mentioned above.

**Other Strengths And Weaknesses:**

Strengths:

1. The paper studies an interesting problem: promoting the positive impact of label information on unlabeled nodes.

2. The proposed method is relatively novel and reasonable.

3. Theoretical analysis enriches the foundation of this work.

Weaknesses:

1. When computing the ELU graph, the paper employs a variant of GCN in the form of Eq.4. However, it remains unclear whether the same form of GCN is also utilized during the contrastive learning phase. This aspect necessitates further clarification.

2. To validate the effectiveness of the ELU graph, the authors should compare it with alternative graph construction methods, such as the KNN-graph, by replacing the ELU graph in the ELU-GCN and reporting the corresponding results.

3. The improvement of NELU nodes after using ELU-GCN should be reported.

**Questions For Authors:**

Why can't the two-stage framework proposed in the paper be designed as an end-to-end framework?

**Relation To Broader Scientific Literature:**

The paper builds upon graph structure learning, further considering and enhancing the influence of label information on unlabeled nodes. Additionally, it introduces the new contrastive learning paradigm to improve the integration of learned graph structures with the original graph information.

**Theoretical Claims:**

The paper provides interesting theoretical claims and the proofs appear sound, but there are still some limitations. For example, the $Y_{true}$ is actually unknown to us, thus this may limit the applicability of the theory.

---

> ### Author Rebuttal · Authors · 2025-03-30
>
> Thanks for the positive comments. We are so encouraged and will try our best to address the concerns one by one. All changes here will be found in the final version.
>
> >Q1: Limitation on theoretical part: the $\mathbf{Y} _{true}$ is actually unknown to us, thus this may limit the applicability of the theory.
>
> A1: It is true that $\mathbf{Y} _{true}$ is unknown to us. However, we can approximate $\mathbf{Y} _{true}$ using the pseudo-labels predicted by GCN, a common practice in many existing works [1]. This approximation allows us to learn an adjacency matrix
>  $\mathbf{A}$ that ensures strong generalization by Theorem 2.3. By then applying GCN to this learned  $\mathbf{A}$, we can obtain higher-quality pseudo-labels that more closely approximate $\mathbf{Y} _{true}$. Through this iterative process, the learned ELU graph progressively refines $\mathbf{A}$ and pseudo-labels, approaching the optimal structure for generalization.
>
> [1] Calibrating graph neural networks from a data-centric perspective. WWW'24
>
> >Q2: When computing the ELU graph, the paper employs a variant of GCN in the form of Eq.4. However, it remains unclear whether the same form of GCN is also utilized during the contrastive learning phase. This aspect necessitates further clarification.
>
> A2: We ultimately use the standard GCN formulation (Kipf \& Welling, 2017) in the contrastive learning phase. The variant of GCN in Eq. 4 is introduced solely to facilitate the optimization of the objective function in Eq. 5.
>
> >Q3: To validate the effectiveness of the ELU graph, the authors should compare it with alternative graph construction methods, such as the KNN-graph, by replacing the ELU graph in the ELU-GCN and reporting the corresponding results.
>
> A3: The effectiveness of the ELU graph is demonstrated by its ability to make unlabeled nodes use the label information more effectively while enhancing generalization. However, the KNN graph only considers the feature information, which ignores the impact of labels. To further validate this,  we have added an additional experiment where the ELU graph in ELU-GCN is replaced with the KNN graph for further validation. The results are shown in the table below.
>
> | Datasets | Cora             | Citeseer         | pubmed           | Computers        | Photo            | Chameleon        | squirrel         |
> |----------|------------------|------------------|------------------|------------------|------------------|------------------|------------------|
> | KNN-GCN  | 82.73±0.64       | 72.15±0.34       | 79.57±0.51       | 82.16±3.52       | 90.75±1.54       | 48.16±3.42       | 36.49±3.56       |
> | ELU-GCN  | ***84.29±0.39*** | ***74.23±0.62*** | ***80.51±0.21*** | ***83.73±2.31*** | ***90.81±1.33*** | ***70.90±1.76*** | ***56.91±1.81*** |
>
> It is evident that replacing the ELU graph with the KNN graph leads to a significant performance drop. This confirms that the ELU graph effectively facilitates the utilization of label information by unlabeled nodes, whereas the KNN graph, which relies solely on feature similarity, fails to capture this crucial aspect.
>
> >Q4: The improvement of NELU nodes after using ELU-GCN should be reported.
>
> A4: We have already conducted this experiment, and the results are presented in Figure 7 of the appendix. This figure illustrates the improvement of NELU nodes after using ELU-GCN.
>
> >Q5: Why can't the two-stage framework proposed in the paper be designed as an end-to-end framework?
>
> A5: While our framework follows a two-stage design, the first stage—constructing the ELU graph—is a parameter-free process. Since this stage does not involve learnable parameters, it cannot be seamlessly integrated into an end-to-end framework. Overall, the ELU graph is firstly precomputed and then used to enhance GCN training in the second stage. This parameter-free approach ensures stability and efficiency in graph construction while keeping the model focused on learning meaningful representations during the second stage. In our future work, we plan to make our framework be an end-to-end one.

---

### Decision · Program_Chairs · 2025-05-01

**Decision:**

Accept (poster)

**Comment:**

This paper proposes a two-step framework. ELU-GCN, to improve the effectiveness of labels. It learns a new graph structure and designs a new graph contrastive learning by exploring the relationship between the learned graph and the original graph. Theoretical and experimental justifications demonstrate its superiority in generalization ability. The claims are supported by clear and convincing evidence. The proposed method makes sense. Reviewers believe the proposed method is novel and that theoretical analysis is solid. After rebuttal, one reviewer increases his ratings. All reviewers agree to this paper.